# Microscale carbon distribution around pores and particulate organic matter varies with soil moisture regime

Steffen Schlüter [1]✉, Frederic Leuther [1], Lukas Albrecht [1], Carmen Hoeschen [2], Rüdiger Kilian [3], Ronny Surey [4], Robert Mikutta [4], Klaus Kaiser [4], Carsten W. Mueller [2,5] & Hans-Jörg Vogel[1,4]

Soil carbon sequestration arises from the interplay of carbon input and stabilization, which vary in space and time. Assessing the resulting microscale carbon distribution in an intact pore space, however, has so far eluded methodological accessibility. Here, we explore the role of soil moisture regimes in shaping microscale carbon gradients by a novel mapping protocol for particulate organic matter and carbon in the soil matrix based on a combination of Osmium staining, X-ray computed tomography, and machine learning. With three different soil types we show that the moisture regime governs C losses from particulate organic matter and the microscale carbon redistribution and stabilization patterns in the soil matrix. Carbon depletion around pores (aperture > 10 μm) occurs in a much larger soil volume (19–74%) than carbon enrichment around particulate organic matter (1%). Thus, interacting microscale processes shaped by the moisture regime are a decisive factor for overall soil carbon persistence.

[1] Department of Soil System Science, Helmholtz-Centre for Environmental Research UFZ, Halle, Germany. [2] Chair of Soil Science, TUM School of Life Sciences, TU Munich, Freising, Germany. [3] Institute of Geoscience and Geography, Martin-Luther-University Halle-Wittenberg, Halle, Germany. [4] Institute of Soil Science and Plant Nutrition, Martin-Luther-University Halle-Wittenberg, Halle, Germany. [5] Department of Geosciences and Natural Resource Management, University of Copenhagen, Copenhagen, Denmark. ✉email: steffen.schlueter@ufz.de

Long-term storage of carbon (C) in the soil is an important contribution to the global C reservoir[1,2] and is at the same time crucial for sustaining essential ecosystem functions[3]. C storage in soil depends on the amount and quality of C inputs and their subsequent stabilization by various mechanisms which vary in time and space[4,5]. The belowground C input by plants through particulate organic matter (POM), especially through fine roots, provides large amounts of biodegradable C that are either immediately lost through microbial respiration or assimilated and subsequently stabilized in the surrounding soil matrix[6], where it fosters the formation of mineral-associated organic matter (MAOM)[7]. Physical protection of organic matter against mineralization by occlusion within pores and organic matter interactions with minerals have been identified as key processes for increasing the persistence of C in soil[8–10]. The soil moisture regime has a strong impact on both processes as high soil moisture levels limit oxygen availability to microorganisms and thus protect soil C from mineralization in anoxic microsites[11–13]. At the same time, high water contents facilitate leaching and diffusive transport of C from POM towards mineral sorption sites[14,15]. At the soil profile to landscape-scale, the strong influence of moisture on C storage manifests itself by the slow organic matter decomposition in permanently wet soils[16,17] and the fast C mineralization with increasing $O_2$ availability[18–20]. Laboratory incubation experiments underpinned the impact of water saturation and oxygen availability on C mineralization[21–23]. They further demonstrated the coexistence of different microbial communities in different pore size classes under different soil moistures and how this impacts mineralization rates[24–26]. However, there is a striking paucity of data about in-situ microscale patterns of C distribution that might arise from different soil moisture regimes.

Based on these macroscopic observations, we hypothesize that soil moisture regimes shape characteristic C distribution patterns at the pore scale. Until recently this hypothesis was impossible to address experimentally as conventional organic matter fractionation methods rely on disrupting the intact soil structure[27]. Mapping of organic matter fractions in intact soil, in turn, is typically restricted to microscopic or spectroscopic techniques on two-dimensional soil sections, a procedure too laborious to represent larger soil volumes[28,29].

Here we introduce a novel combination of C staining using osmium tetroxide ($OsO_4$), three-dimensional Os mapping with polychromatic X-ray computed tomography (X-ray CT), and detection of different organic matter fractions in undisturbed soil samples using machine learning. We show that these image-derived organic matter fractions accurately reflect C fractions obtained by conventional physical soil fractionation, such as POM, water-extractable organic carbon (WEOC), and to a lesser degree also MAOM, with the added benefit of quantifying their position relative to the undisturbed pore space. This, for the first time, allows for the exact mapping of C concentration gradients in the direct vicinity of pores and POM particles. In line with classical 2D micro-morphological observations on soil sections[30] and validated by independent determination of conventional soil fractions and short-term soil incubations, POM was differentiated into fibrous and compact POM based on 3D morphological properties. Fibrous, elongated shapes, represent fresh litter and roots that are mostly located inside the intact pore system. Compact POM, with a sizeable share of biochar, has a lower surface-to-volume ratio with fewer internal voids and more rounded shapes, as a result of repeated fragmentation and translocation. Using incubations of soil aggregates under different oxygen and moisture conditions we show that C mineralization related well to the image-derived amount of C stored in these POM types. While the estimation of C stored in POM is accurate, we demonstrate with correlative microscopy employing microscale X-ray fluorescence microscopy (µXRF) and nanoscale secondary ion mass spectroscopy (NanoSIMS) that MAOM contents cannot be reliably assessed, because $OsO_4$ also sorbs to reactive soil minerals. Relative changes in average Os sorption as a function of pore or POM distances can still be interpreted as spatial gradients in sorption to organic matter assuming that Os sorption to mineral surfaces is random with respect to these distances. Based on these methodological advancements we demonstrate that three soils with contrasting soil moisture regimes, a well-aerated regime under dry climatic conditions (Haplic Chernozem), an alternating-wet, slowly draining regime (Stagnic Luvisol), and a permanently wet, groundwater-affected regime (Fluvic Gleysol), are all depleted in C in the direct vicinity of pores, presumably due to more fluctuating microenvironmental conditions that promote C mineralization by locally enhanced oxygen availability and enhanced C desorption due to equilibration with the more frequently exchanged soil solution. We also show that C enrichment occurs in the soil matrix around fibrous and compact POM in well-aerated and stagnant soils (Chernozem and Luvisol), but not in a groundwater-affected Gleysol. Consequently, these POM types encounter completely different microenvironments and leaching histories under differing soil moisture regimes, with immediate consequences for C turnover as well as MAOM formation in the surrounding soil matrix.

## Results

**C mapping in soil with correlative imaging.** Fine-textured topsoils were sampled at three sites in Germany managed either as long-term grassland or cropland. The soils are characterized by different mineralogy, pedogenesis, and soil moisture regimes. They vary strongly in organic C contents as well as in proportions of POM and MAOM (Supplementary Table 1) due to different parent materials and land use. Therefore, each soil is supposed to feature a different contribution to governing C stabilization mechanisms. Large soil aggregates (4–8 mm) were collected by sieving and then subjected to C staining by $OsO_4$ vapor in a dry state. Osmium is reported to selectively bind to olefinic double bonds present in organic compounds such as lipids, amino, and fatty acids[31,32]. For the first time, Os intensity was not mapped by synchrotron-based X-ray CT dual-energy scanning, but by subtracting spatially aligned, consecutive scans acquired with a commercial, polychromatic X-ray CT scanner before and after Os staining (Supplementary Fig. 1). With 3D image processing involving machine learning-based image segmentation, the microstructure was classified as pores, POM, matrix, and dense areas, such as sand grains and iron (Fe) nodules (Fig. 1a, b, 5 µm voxel resolution). In addition, POM was further segmented into fibrous and compact POM as described above (Fig. 1d, Supplementary Fig. 2).

All soils exhibited a rather uniform spatial distribution of Os in the matrix of soil aggregates (Fig. 1c, displayed for all 24 soil aggregates in Supplementary Fig. 3), with the expected difference in magnitude, reflecting the different MAOM levels of the studied soils (Supplementary Table 1). Moreover, characteristic differences between the soils emerged with respect to Os concentrations in areas classified as POM and dense areas. In soil aggregates of the Chernozem and Gleysol, these dense areas were mainly composed of solid particles impermeable to $OsO_4$ vapor and therefore remained unstained. In soil aggregates of the Luvisol, dense areas constituted porous and highly Os-enriched nodules rich in Fe that formed under hydromorphic conditions[33,34]. POM occasionally had the highest Os intensities of all material classes (Fig. 1c) but Os enrichment in individual

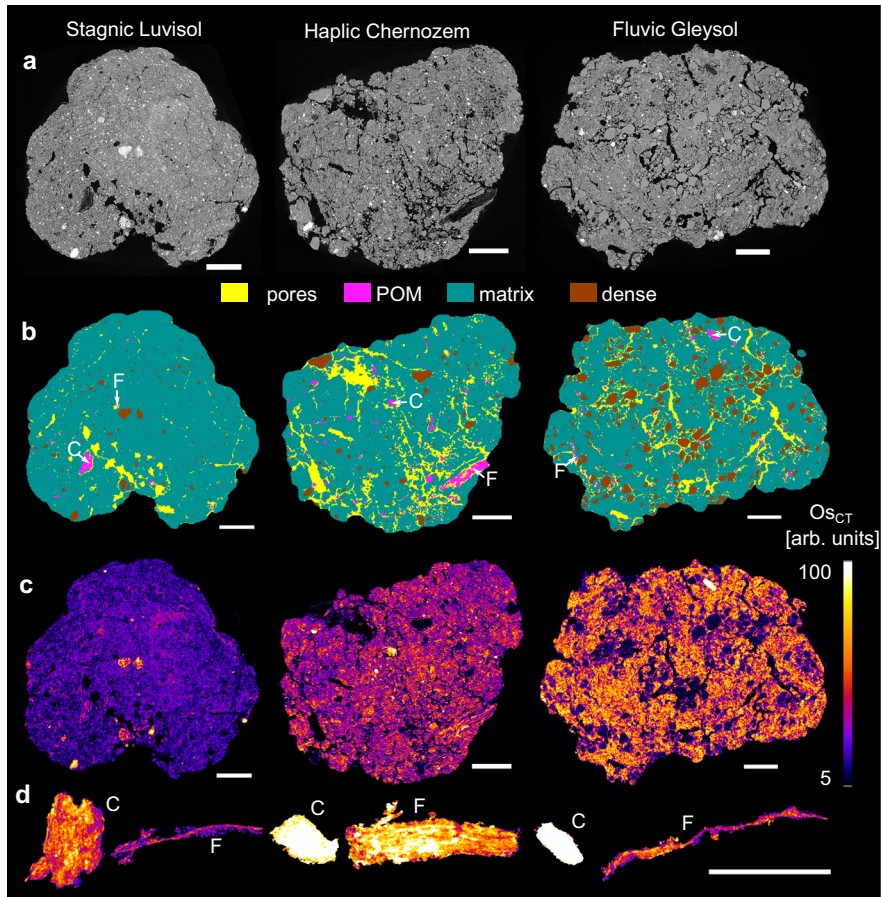

**Fig. 1 Microscale characterization of soil structure. a** Two-dimensional slices of X-ray tomograms for selected soil aggregates of the Stagnic Luvisol, Haplic Chernozem, and Fluvic Gleysol prior to Os staining. **b** Segmented images showing the spatial distribution of pores, POM, soil matrix, and dense areas, including rock fragments and Fe-rich nodules. Particulate organic matter is segmented into compact POM (C) and fibrous POM (F) for which one example is shown for each soil. **c** Difference images of the same slices displaying organic matter stained by Os. The Os concentration (Os$_{CT}$) is normalized by reference materials and has arbitrary units. **d** The average Os concentration in each C and F example is shown as 2D maximum intensity projection through the 3D object. Scale bar represents 1 mm.

POM areas varied tremendously with the chemical composition and inner surface area of the POM (Fig. 1d).

The quantification of C via Os mapping with polychromatic X-ray CT was in line with total organic C contents (TOC) and organic C fractions (MAOM, POM, WEOC) independently determined on separate sets of aggregates and also with registered Os maps acquired with μXRF on several soil sections per aggregate (Fig. 2). The agreement between the average Os intensity detected with μXRF (Os$_{XRF}$) and TOC was excellent ($R = 1.00$, $p = 0.042$, Fig. 2a). Correlative microscopy revealed the congruency of spatial Os patterns detected with X-ray CT and μXRF with only minor differences due to different depth resolutions (Supplementary Fig. 4). Yet, only X-ray CT allowed for morphological separation into POM and matrix-bound organic matter. The accordance of X-ray CT-derived amounts of matrix-bound Os (Os$_{CT}$ intensity in matrix × volume fraction of matrix) and conventionally determined MAOM contents were somewhat weaker ($R = 0.91$, $p = 0.28$) than for TOC, likely due to unresolved POM with high Os$_{CT}$ intensity in the Chernozem soil matrix (Fig. 2b). The image-derived POM amount, i.e., the amount of Os bound to resolved POM (Os$_{CT}$ intensity in POM × volume fraction of POM), correlated well ($R = 0.98$, $p = 0.11$) with the independently determined POM content (Fig. 2c). The share of C in fibrous POM, i.e., the ratio of Os bound to fibrous POM over Os in all resolved POM, adequately reflected ($R = 1.00$,

$p < 0.001$) the share of organic C that was extracted from soil aggregates by water (Fig. 2d). No such well-matching relationship could be obtained for organic matter fractions from conventional density separation (Supplementary Table 1).

Noteworthy, regressions for MAOM and TOC vs. image-derived Os amounts had non-zero intercepts (Fig. 2a, b), suggesting that there was some adsorption of Os to mineral surfaces. The sorption to minerals was confirmed by Os intensities in independently scanned reference materials that were subjected to the same staining protocol (Fig. 3). Although there was no Os adsorption to non-porous materials and to quartz silt with unresolved pores, the adsorption to goethite (α-FeOOH) and illite (2:1 clay mineral) was similar to some reference char and plant materials within inherent porosity. The variability in Os sorption to POM references (Fig. 3) was also in line with the observed variability in Os sorption to POM within the studied soils (Fig. 1d).

Elemental mapping with μXRF revealed co-localization of Os with other elements, e.g., phosphorus (P) or Fe that may indicate Os-binding sites (Supplementary Fig. 5). Yet, the technique was unsuitable to distinguish whether Os was sorbed to minerals or organic molecules. This was better achieved with NanoSIMS imaging of selected spots within different micro-environments previously identified by X-ray CT and μXRF (Fig. 4). NanoSIMS analysis demonstrated that the highest Os occurrences were co-localized with $^{12}C^{14}N$ ions in the cell walls

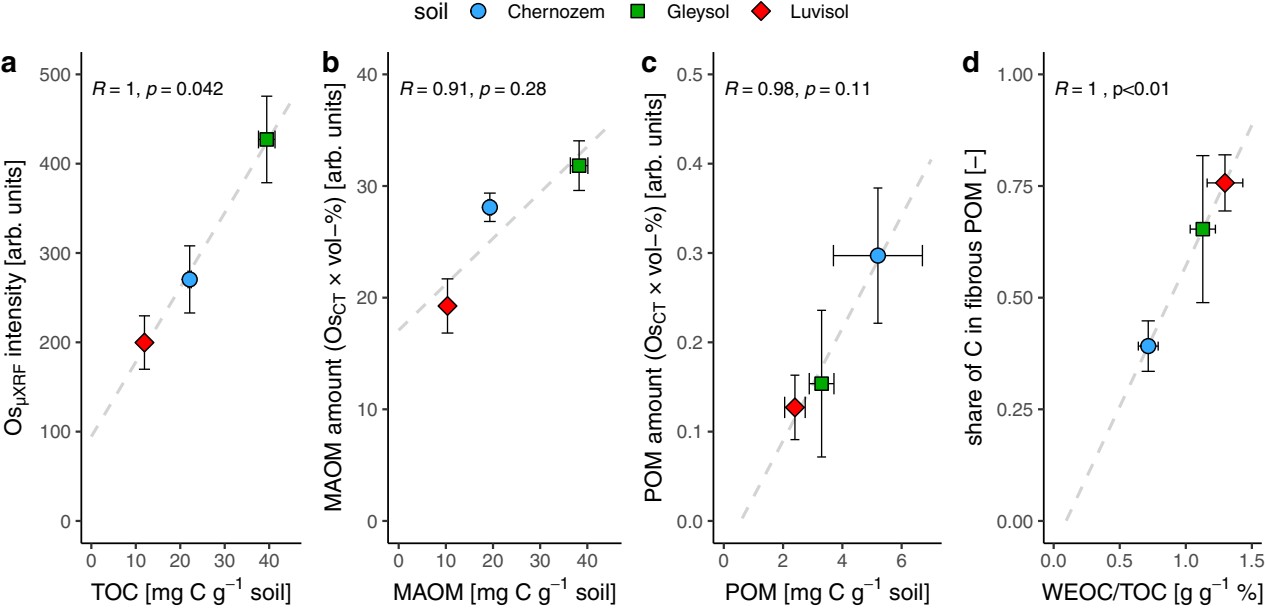

**Fig. 2 Comparison of C in organic matter fractions derived from the novel imaging protocol and conventional methods. a** Average Os intensity detected with µXRF (Os$_{XRF}$) on soil sections as related to total organic C content (TOC). **b** Image-derived mount of mineral-associated organic matter (MAOM) derived from the product of average Os intensity of matrix voxels and the volume fraction of matrix voxels over all soil voxels as related to the soils' contents of C within MAOM. **c** Same comparison as in **b** but for C in particulate organic matter (POM). **d** Image-derived share of C bound to fibrous POM over C bound to total POM as related to the fraction of water-extractable organic C (WEOC/TOC) in soil aggregates. Arbitrary units in Os intensity arise from normalization with different references materials. Error bars correspond to two standard errors in each direction with $n = 8$ for image-derived data (y axis) and $n = 3$ for conventional analyses (x axis). All linear regressions are based on $n = 3$.

of plant residues (Fig. 4, spot 7−9). In the soil matrix (Fig. 4, spot 4−6) and Fe-rich nodules (Fig. 4, spot 1−3), the spatial distribution of Os suggested binding to Fe-bearing minerals ($^{56}Fe^{16}O^-$ ions) and clay minerals ($^{27}Al^{16}O^-$ ions) in addition to sorption to organic matter patches, and thus, confirmed the findings for Os sorption to reference materials (Fig. 3). Regression analysis showed that variation in $^{192}Os^-$ intensities were best explained by $^{12}C^{14}N^-$ ions in plant residues, whereas $^{27}Al^{16}O^-$ ions were a better predictor in nodules and both perform equally in the soil matrix (Supplementary Fig. 6).

Our correlative imaging approach implies that absolute values of local Os concentrations cannot reliably be used for quantification of MAOM in the soil matrix, as there is always some contribution of Os adsorbed to reactive minerals and unresolved POM. In the following, we will therefore designate the matrix Os signals generally as "matrix-bound organic matter" and "C in the soil matrix". We will discuss below how far average matrix-bound C contents can still be investigated with respect to pore or POM distances.

**Microscale C distribution in different soil types**. Individual aggregates from the same soil varied considerably in their average Os concentration due to the variability of organic C contents in the field (Supplementary Fig. 3). Interestingly, there was a linear relationship between increasing matrix Os contents detected by X-ray CT (Os$_{CT}$) and decreasing matrix gray values prior to Os staining (Fig. 5a). The average Os$_{CT}$ intensity of POM seemed much lower in the alternating-wet Stagnic Luvisol and permanently wet Fluvic Gleysol than in the dry Haplic Chernozem (Fig. 5b). The non-zero Os$_{CT}$ in the dense material class of all soils was partly due to shortcomings of image processing (matrix wrongly assigned to dense material) and partly due to minerals and nodules that adsorbed Os, in particular in the Stagnic Luvisol.

So far only differences in average Os concentrations between soil types and material classes have been assessed. Deeper insights into the role of microenvironments as drivers for C sequestration can only be achieved by quantifying the spatial Os distribution. Pore distances had a dominant effect on Os$_{CT}$ intensity gradients within the soil matrix, as they decreased markedly towards pore boundaries (Fig. 5c). Even in the well-aerated Chernozem there was a 30 µm thick Os$_{CT}$ depletion zone around visible pores. The spatial extent of the depletion zone was similar in the poorly drained Luvisol. The spatial extent of the depletion zone was largest in the Gleysol, amounting to 60−80 µm and gradually extending beyond >100 µm. The volume fraction of the soil matrix affected by Os depletion around visible pores amounted to 19% (Luvisol), 65% (Chernozem), and 74% (Gleysol), respectively, because of different visible porosities and internal pore surface area densities (Supplementary Table 1).

Also, the Os concentration in POM was reduced in close vicinity to pores (<20 µm) for all three soils (Fig. 5d). At greater distances to pores (>20 µm), the Os concentration was either increasing (Chernozem), constant (Luvisol), or decreasing (Gleysol) because the proportion, average pore distances, and susceptibility to Os staining of fibrous and compact POM differed between soils.

All segmented POM particles large enough for shape assessment (5149 individual particles) were assigned to either compact or fibrous POM (depicted for all 24 samples in Supplementary Fig. 2). Every POM particle was then attributed with an average Os concentration and an average distance to the next visible pore (Fig. 6b). This pore distance was surprisingly short irrespective of aggregate density and visible porosity and amounted to 21 µm when averaged across all soils and POM types (Supplementary Table 1). It was substantially shorter than the average pore distance in the soil matrix, which was in the range of 45–106 µm depending on visible porosity (Fig. 6b, Supplementary Table 1). The distance

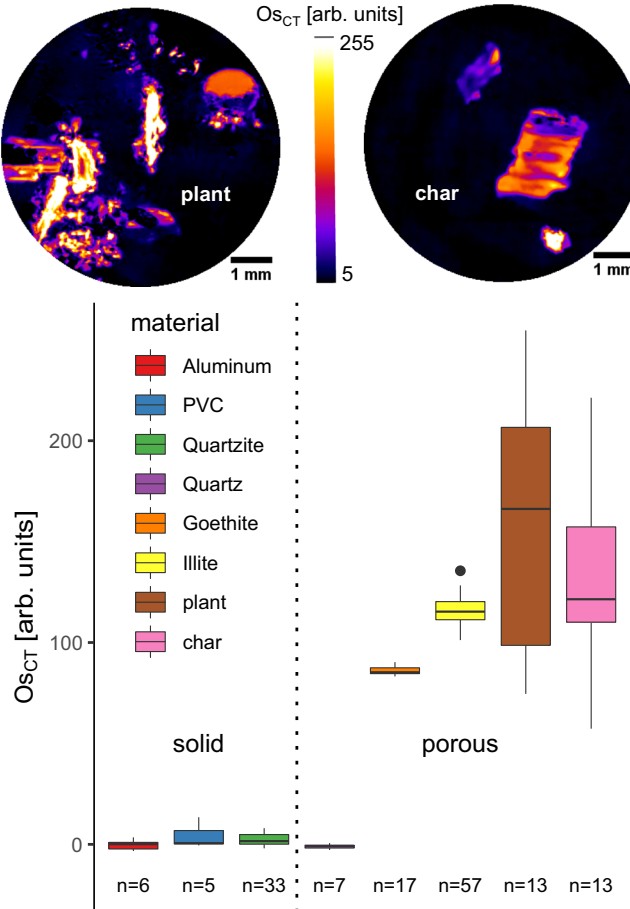

**Fig. 3 Normalized Osmium concentrations (Os$_{CT}$) in some reference materials in arbitrary units.** Some materials are solid and cannot be penetrated by Os vapor. The divergence from zero Os intensity for solid materials can therefore only be explained by polychromatic X-ray artifacts. Other materials are porous, i.e. powders with grain size smaller than the image resolution or biological material within inherent porosity. The boxplots show the 0%, 25%, 50%, 75, and 100% percentiles of varying sample numbers (n) after outlier detection. The two images show representative slices through X-ray tomograms of char and plant samples.

between POM and pores increased to 120 μm when only connected pores were considered for distance estimation (Supplementary Fig. 7), i.e., neglecting isolated pores and only considering pores with continuous access to aggregate boundaries. This is a situation more likely to be encountered for air-filled pores, when soil is drained from full saturation to field capacity.

Moreover, compact POM had a significantly ($p < 0.001$) larger pore distance than fibrous POM (Fig. 6a) in all three soils. Surprisingly, the average pore distance of both POM types did not depend on soil type. Only when the filling of pores by air and water is explicitly considered at field capacity, did the higher aggregate density and lower visible porosity (Supplementary Table 1) also entail significantly higher POM distances to air-filled pores in the Luvisol ($p < 0.001$) (Supplementary Fig. 7). Osmium intensities in POM differed significantly between soil types ($p < 0.001$), but there was no consistent pattern as to whether fibrous or compact POM featured higher Os intensities (Fig. 6b). The interaction term between soil type and POM type was significant ($p < 0.01$). That is, in the well-aerated Chernozem, compact POM was stained more intensively, whereas fibrous POM was stained more intensively in the hydromorphic Gleysol and Luvisol.

The soil matrix around POM was enriched in Os with a spatial extent of $30-40$ μm for the Luvisol and Chernozem soil, irrespective of POM type (Fig. 6b, c). The volume fraction affected by the enrichment amounted to 1.3% in both soils, which was much smaller than the volume fraction affected by Os depletion around pores (Chernozem: 65%, Luvisol: 19%, Supplementary Table 1). A remarkable exception was the groundwater-affected Gleysol, where Os enrichment around POM was completely absent. Strikingly, the soil matrix around fibrous POM embedded in a well-connected, aerated pore network was even depleted in Os (Fig. 6e), whereas there was less Os depletion around compact POM (Fig. 6d) that is on average farther away from pores (Fig. 6a, Supplementary Fig. 7). The volume fraction of the soil matrix affected by Os depletion around fresh POM in the Gleysol was again much smaller (2.8%) and mainly included in the matrix volume affected by Os depletion around pores (74%, Supplementary Table 1).

**C turnover under different moisture regimes**. We speculate that these insights from microscale imaging do not only reflect the long-term fate of organic C in soil, but have also implications for short-term C turnover. To test this hypothesis, sets of individual aggregates were brought to different moisture levels (partially wet vs. fully saturated) and incubated under different oxygen availabilities (20% $O_2$ vs. 0% $O_2$) to measure $CO_2$ efflux rates for 3 days.

The order of C mineralization rates was the same for all boundary conditions (Chernozem<Luvisol<Gleysol, Fig. 7a), but different from the order of TOC contents. This is because the Chernozem evoked exceptionally low $CO_2$ efflux despite the highest POM content (Supplementary Table 1). Under completely anoxic conditions without local differences in metabolic pathways by varying oxygen availability, the different order and even the magnitude of mineralized C was well related ($R = 0.99$, $p = 0.087$) to the fraction of C stored in fibrous POM as analyzed by X-ray CT (Fig. 7b). Consequently, the small fraction of mineralized C in the Chernozem resulted from the high proportion of C stored in compact POM, also comprising biochar.

The highest C mineralization rates occurred at fully water-saturated, oxic conditions in all soils (Fig. 7a), and dropped by a factor of 3–4 under anoxic conditions. The magnitude of $CO_2$ efflux reduction under fully anoxic conditions is closely related ($R = 1.00$, $p = 0.028$, Fig. 7c) to the WEOC content of POM. In other words, readily soluble C from POM seemed to fuel oxic respiration and evoke microbial activity more than other organic matter fractions, but its biodegradability under anoxic conditions also suffered more relative to other fractions.

The reduction in C mineralization towards drier, oxic conditions indicated that with decreasing water content the decreasing diffusion of C to decomposers outweighed the gradual improvement in oxygen availability. That relative $CO_2$ efflux reduction towards drier conditions scaled linearly with the image-derived volume fraction of fibrous POM in soil aggregates ($R = 0.98$, $p = 0.12$, Supplementary Fig. 8).

**Discussion**

The good agreement between image-derived POM amount and conventional POM contents (Fig. 2c) confirmed that Os staining is very accurate in detecting C in organic soil residues[35]. However, a thorough validation of our protocol for microscale C mapping in the soil matrix revealed mixed results as to whether Os intensities can be interpreted towards MAOM. On the one hand, the order of average matrix Os intensities was the same as the order of conventionally determined MAOM contents for all

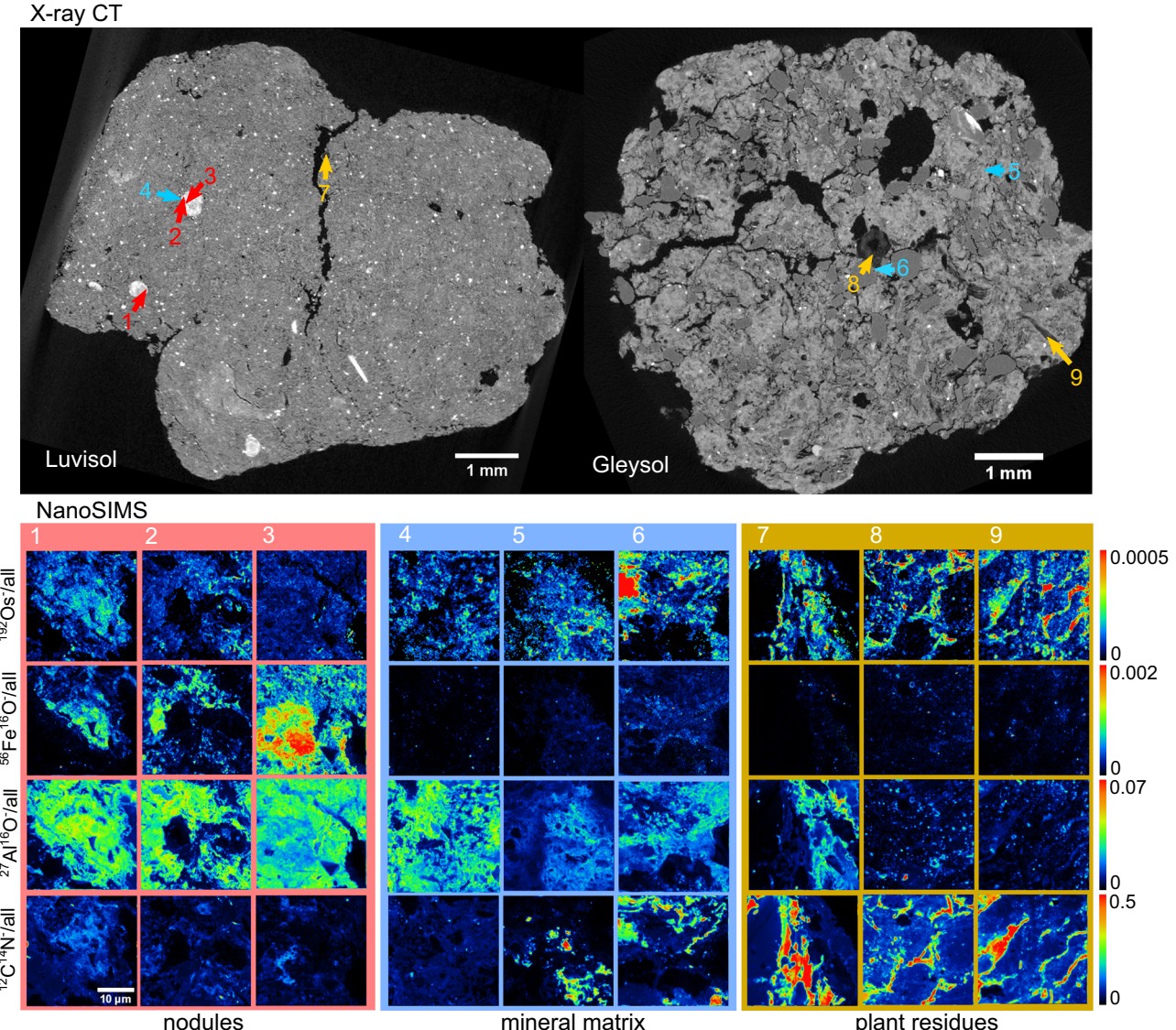

**Fig. 4 Correlative microscopy reveals Osmium sorption mechanisms.** Two-dimensional cross-sections through tomograms (X-ray CT) of aggregates from a Stagnic Luvisol and a Fluvic Gleysol scanned after Os staining. Nine out of 15 spots mapped with secondary ion mass spectroscopy (NanoSIMS) across both soil sections were selected to depict the spatial distribution of organic C ($^{12}C^{14}N^-$/all) and Os ($^{192}Os^-$/all) and its co-localization with Fe ($^{56}Fe^{16}O^-$/all) and Al ($^{27}Al^{16}O^-$/all) in different microenvironments (Fe-rich nodules (red), mineral matrix (blue), plant residues (gold)).

three soils (Fig. 2b). Moreover, the inverse relationship between matrix Os intensities and matrix gray values prior to Os staining (Fig. 5a) is in line with the well-known decrease in bulk density[36], increase in field capacity[37], and more specifically the increase in the sub-resolution porosity[38] with increasing C contents. On the other hand, matrix Os intensities were modulated by varying contents and staining intensities of unresolved POM particles (Fig. 2b). In addition, as shown by regression analysis (Fig. 2a, b), staining of reference materials (Fig. 3), and NanoSIMS imaging (Fig. 4), there was a substantial contribution of reactive minerals to Os sorption in the soil matrix. Such Os sorption to the mineral matrix, though smaller than to POM, was also reported for an allophanic Andosol[39]. Therefore, the current assertions of Os selectively binding to double bonds in organic matter[35,38–41] and therefore only staining organic molecules without adsorption to mineral surfaces need to be reconsidered. A range of bonding mechanisms to reactive surfaces was suggested in the past[42], including hydrogen bonds[43]. For these reasons, matrix Os intensities should not be equated to MAOM contents, especially

considering comparisons across soils with different parent materials, unless some correction is employed that has yet to be developed. However, relative changes in average matrix-bound organic matter contents with respect to pore or POM distances can still be investigated assuming that mineralogy does not change systematically at these distances.

Violations of the assumed indifference to pore distances can be easily imagined, especially in redoximorphic soils, where the mobility of Fe depends on the local oxidation state[33,34,44]. In fact, μXRF maps demonstrated that there is indeed a decrease in Al and Fe as proxies for adsorbing minerals towards pores in hydromorphic soils (Supplementary Fig. 9). The depletion of Al around pores may hint toward dispersive clay mobilization around pores. Yet, the spatial extent of both the Fe and Al depletion zone was much larger, amounting to ~200 μm and 600 μm in the Stagnic Luvisol and Fluvial Gleysol, respectively. Moreover, there was no Al and Fe depletion around pores in the Haplic Chernozem despite the Os depletion detected with X-ray CT. Also, sub-resolution porosity gradients were ruled out

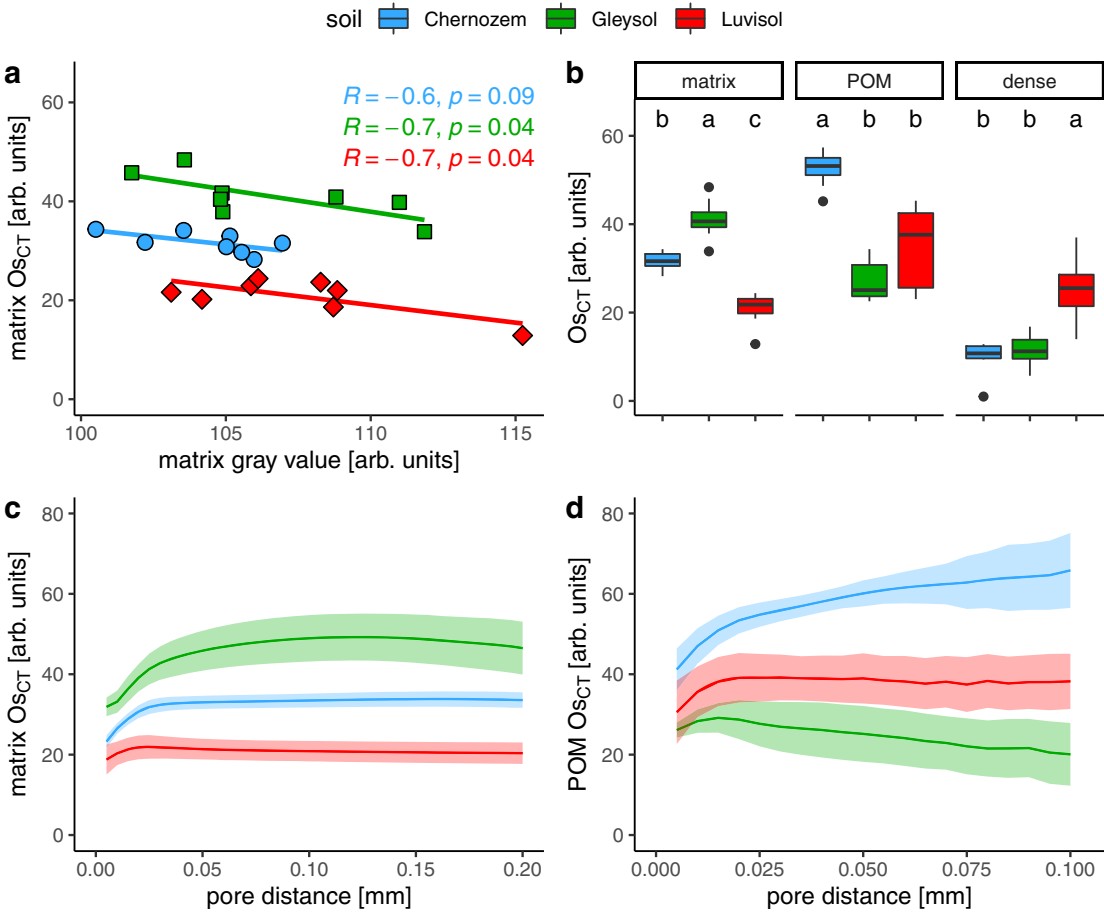

**Fig. 5 Osmium concentrations (Os_{CT}) with respect to material classes and pore distances. a** Linear relationship between Os concentration in the soil matrix as a proxy for mineral-associated organic matter and the soil matrix gray value prior to Os staining as proxy for bulk density. Both properties are given in arbitrary units resulting from grayscale normalization. **b** Average Os concentration in different material classes of each soil. The boxplots show the 0%, 25%, 50%, 75%, and 100% percentiles after outlier detection ($n = 8$). Osmium concentrations in the soil matrix **c** and in POM **d** decline towards pore boundaries. Shaded areas represent two standard errors in each direction ($n = 8$).

previously as an explanation for lower Os_{CT} around pores in a similar study employing synchrotron X-ray CT scans at two resolutions (13 μm vs. 2 μm)[38]. This leaves a decrease in MAOM content or less efficient Os staining of MAOM due to altered chemical composition around pores as most likely explanations for the observed pattern that cannot be disentangled by the applied methods.

In summary, Os mapping with polychromatic X-ray CT is a viable alternative to synchrotron X-ray CT since both the magnitude and spatial distribution of Os are well in line with conventional physical organic matter fractions and μXRF analyses. Our novel image processing protocol opens up new opportunities to analyze more and larger samples unconstrained by the limited field of view and limited access to synchrotron-based CT.

Our findings indicate that soil moisture regimes and the position of organic matter within the pore space shape the C distribution at the microscale through the interplay between C losses from POM and C redistribution and stabilization in the soil matrix imposed by the pore network as conceptualized in Fig. 8. POM within the pore network is an important source for the formation of MAOM, either in its immediate vicinity or even in the distant soil matrix, depending on the soil moisture regime. The magnitude and range of POM leaching, i.e., the translocation of soluble C compounds from POM into its surrounding, varied with soil moisture regime. In the well-aerated Chernozem under

dry climate conditions, C loss from POM was low (Fig. 5b and Fig. 6c) and resulted in a thin layer (30–40 μm) of C enrichment in the adjacent soil matrix (Fig. 6d, e), corroborating the notion of POM as a functional component for MAOM formation by releasing sorptive compounds[7]. Active C translocation by fungal growth rather than diffusion might be the dominant process for this C enrichment especially in well-aerated soils[15,45,46]. The layer is thinner than the 100−130 μm zone of Os enrichment reported previously for a small number ($n = 6$) of POM-bearing pores in a well-drained, loamy soil[38]. In the alternating-wet Luvisol with twice the annual precipitation, POM was more effectively leached (Figs. 5b and 6c), but the released C did not become stabilized farther in the soil matrix (Fig. 6d, e), presumably due to periodic dry spells that prevented diffusive or convective migration of C from POM into the soil matrix. In the groundwater-affected Gleysol, C leached from POM might have migrated farther into the soil matrix and reached more distant sorption sites. The enhanced transport into and stabilization at distant locations prevented the formation of distinct MAOM layers around POM as observed for the other two soils (Fig. 6d, e). The soil moisture regime may not only govern the magnitude of POM leaching, but also the magnitude of POM mineralization, as the C mineralization is impaired by dryness[21,22] that is more common in the Haplic Chernozem. Different degrees of C losses from POM due to leaching or mineralization in the different soil types also

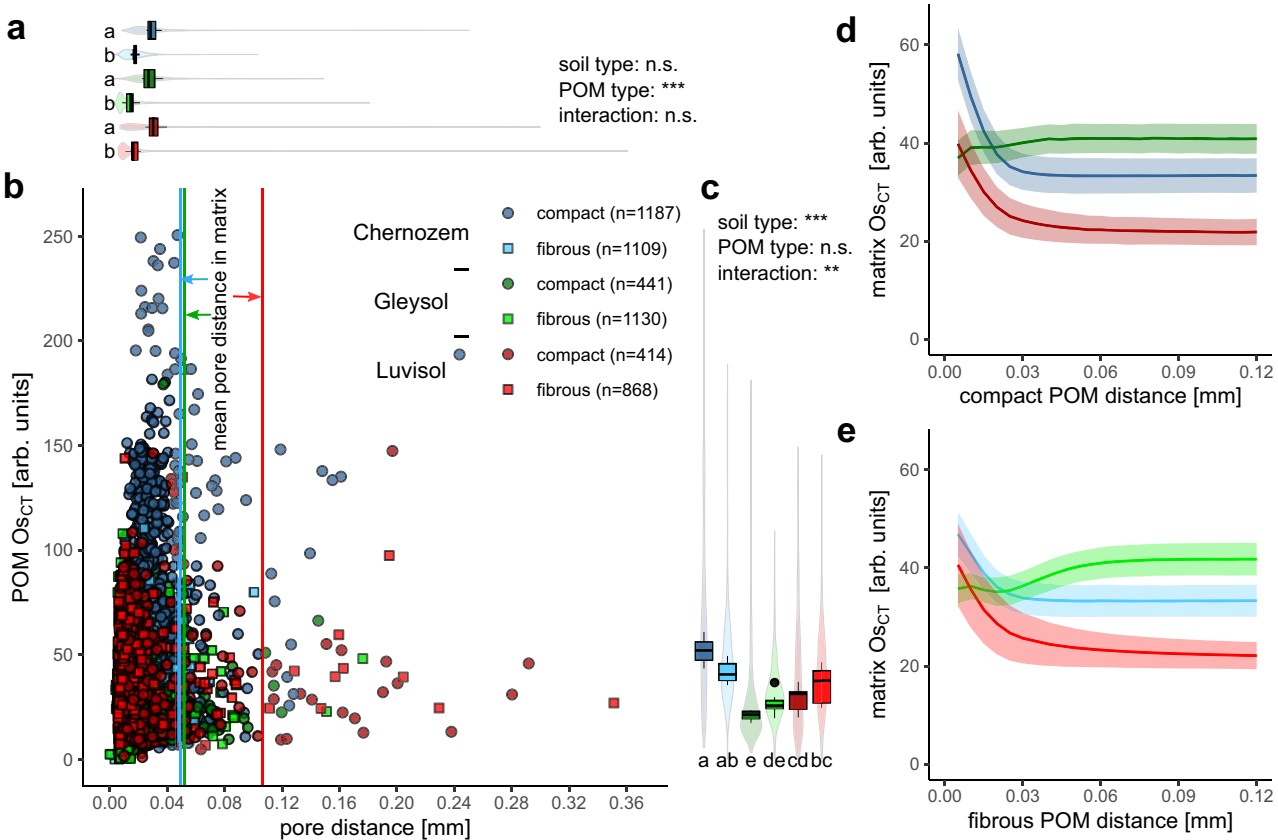

**Fig. 6 Characterization of fibrous and compact particulate organic matter in different soils. b** Average Os concentration ($Os_{CT}$) and average pore distance for individual, fibrous, or compact POM objects. **a** and **c** Violin plots represent marginal distributions of individual POM objects. Boxplots represent the 0%, 25%, 50%, 75, and 100% percentiles of averages for individual aggregates ($n = 8$) with small letters reflecting significant differences tested at $p < 0.05$ (n.s. not significant). **d** and **e** Average Os concentration in the soil matrix around compact and fibrous POM. Shaded areas represent two standard errors in each direction ($n = 8$).

manifested themselves in the interaction effect of soil type and POM type (Fig. 6c). In hydromorphic soils compact POM had lower C contents than fibrous POM due to greater cumulative exposure to soil moisture. This difference in cumulative C depletion and the different degrees of occlusion support previous findings[30] that a morphological POM classification into fibrous and compact can be interpreted towards decomposition stages like "fresh" and "decomposed" (Fig. 8). In the well-aerated Haplic Chernozem the order was reversed. Here, recalcitrant biochar made up a large fraction of decomposed POM. Though being more occluded and likely residing in soil for much longer it featured high Os intensities, indicating that it was less affected by exposure to soil moisture than decomposed POM in hydro-morphic soils.

In contrast to strongly differing C patterns in the soil matrix around POM in response to different soil moisture regimes, we observed depletion of matrix-bound organic matter around pores irrespective of the soil moisture regime (Fig. 5c). The spatial extent of the depletion zone in the Chernozem and Luvisol was similar to a 30−50 μm depletion zone reported for a well-drained loamy soil[38]. In the Gleysol, it mostly ranged between 60 and 80 μm and gradually extended beyond 100 μm, as unresolved pores remained predominantly water-filled throughout the year and aeration patterns were tied stronger to the visible pore space. The C depletion around pores suggests that matrix-bound organic matter in close contact with visible pores (aperture >10 μm), either becomes more easily desorbed due to equilibration with the more frequently exchanged soil solution, encounters

less reactive minerals as potential sorption sites, or is more sus-ceptible to microbial processing due to better aeration which may also alter the chemical composition resulting in less olefinic double bonds. Especially in the hydromorphic soils, all of these processes could be interlinked, since C mobilization is not only promoted by Mn and Fe dissolution at low redox potential[47], but even more so by the concomitant pH increase which favors C desorption from mineral surfaces[48]. In intact hydromorphic soils, microbial processing of labile C along pores might therefore induce anoxic microsites[49] during wet periods and promote C desorption locally, whereas microbial processing along pores promotes C mineralization when they fall dry. Consequently, C stabilized in micro- or mesopores (aperture <10 μm) of the dis-tant soil matrix will possess a higher likelihood to survive for longer time than MAOM or POM located around or in larger pores, respectively. The volume fraction of the distant soil matrix, in which C can potentially be stabilized for centuries to millennia[50,51], ranged from 26% (Gleysol) to 81% (Luvisol) and was mainly governed by bulk density and the soil moisture regime. Our data provide a hitherto unavailable quantitative assessment of the resulting C accumulation pattern, and thus, strengthen the view that various C (de)stabilization mechanisms follow a spatial pattern depending on the structure of the pore network.

Our findings reconcile the contrasting views on whether POM-bearing pores predominantly foster C mineralization[49] or C sequestration[6]. Pores bearing POM, in particular those formed around fine roots (30–150 μm diameter), may act as entry paths

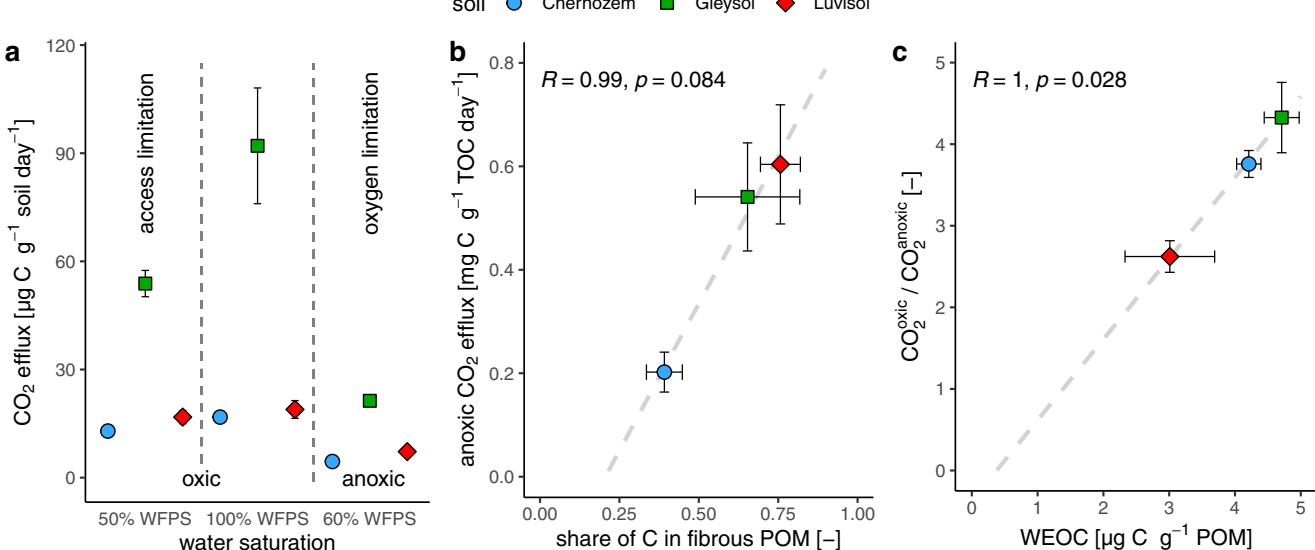

**Fig. 7 Soil respiration results from short-term incubations. a** Absolute $CO_2$ efflux rates under oxic (20% $O_2$) and anoxic (0% $O_2$) conditions at different moisture levels (WFPS, water-filled pore space). **b** Relative $CO_2$ efflux rates, i.e. as a fraction of total organic C, scale linearly with the share of image-derived C in fibrous particulate organic matter (POM) to that in total POM. **c** The increase in $CO_2$ efflux rates through the presence of oxygen (at 100% WFPS) scales linearly with the water-extractable organic C (WEOC) leached from POM. Bars represent two standard errors in each direction with $n = 3$ for $CO_2$ and WEOC data and $n = 8$ for image-derived POM data.

for C into the soil matrix and trigger C accrual in its surrounding, especially in well-aerated soils, as posited by ref. [6]. However, only a small fraction of pores in that size range is actually filled with fresh plant residues[52], amounting to fractions of 5–13% of all visible pores (aperture >10 μm) in the present study. Also, the moisture exposure will govern the spatial extent and probably also the longevity of MAOM enrichment in the soil matrix surrounding pores that contain POM. At the same time, all pores, including those occupied by plant residues, serve as entry paths for oxygen, probably promoting mineralization instead of stabilization of C[49]. In addition, other processes, such as the absence of sorbing minerals, frequent MAOM desorption, and migration of soluble C into the soil matrix, may contribute to C depletion around larger pores. Overall, our data suggest that visible pores holding organic residues may support C formation in the surrounding soil matrix, while empty pores promote C depletion. However, the volume of influence of C depletion around pores is always greater (19–74%) than that of MAOM enrichment around POM (1%), simply because of vastly different volume fractions of pores and POM.

Interestingly, the spatial extent of C enrichment around POM and C depletion around pores was very similar, irrespective of the soil moisture regime. It was 30–40 μm and only extended to 80 μm around pores in the Gleysol, probably because the investigated topsoils had been exposed to repeated tillage (Chernozem, Luvisol) or bioturbation (Gleysol). Soils with slower structure turnover, and thus more static pore networks, such as subsoils, are expected to exhibit different MAOM gradient extents around pores and POM. In addition, we cannot completely rule out confounding effects of soil moisture regime and land use at the selected sampling sites. The two tilled cropland soils were more similar with respect to Os gradients around pores and POM than the non-tilled grassland soil. It is conceivable that soil aggregates remaining intact during aggregate fractionation also remain intact during plowing. Pores bearing POM in such aggregates may therefore have experienced many cycles of C enrichment. Bioturbation in the non-tilled grassland might in turn have led to a more effective homogenization of the soil structure so that POM-

bearing pores are younger. In the future, such interactions between soil type and land use can be scrutinized with a full factorial experimental design comprising all combinations of soil moisture regimes and land uses in controlled long-term trials. The presented C mapping technique paves the way for a comprehensive assessment of these microscale patterns in different pedogenetic settings.

The soil moisture regime exerted a strong influence on the C losses from POM and on the microscale distribution of matrix-bound organic matter. However, the fate of organic C in the soil is not only governed by spatially varying redistribution and stabilization mechanisms but also by how new C enters the soil in the first place[6,53]. Litter and root residues are the main sources of fresh POM, which in turn provides most of the C that become stabilized in the soil matrix. The amount and quality of POM input into the soil are controlled by other factors than soil moisture regime, such as nutrient availability, crop productivity, plant species, and soil management.

Such diverse C inputs have ramifications for the abundance, quality, and spatial distribution of POM at the mm scale and eventually how effectively C is mineralized under different boundary conditions as conceptualized in Fig. 8. We tested the bioavailability and biodegradability at different oxygen and water saturation levels. Bioavailability describes the potential of microorganisms to interact with organic compounds, which may be restricted by sorption, pore size, or dryness[54]. Biodegradability is a measure of the utilization of organic compounds and varies with the intrinsic quality and soil conditions[54]. The biodegradation in terms of the fraction of TOC mineralized per day was not controlled by TOC or the total POM amount, but linked to the image-derived share of C within fibrous POM (Fig. 7b), which in turn was in excellent agreement with the water-extractable fraction of TOC (WEOC/TOC, Fig. 2d). This resulted in the lowest biodegradation of C in the Haplic Chernozem having the highest share of C in recalcitrant biochar. This is in line with previous findings of the vastly different effects of litter and char amendments on soil respiration irrespective of the oxygen availability[55]. It also straightly points at

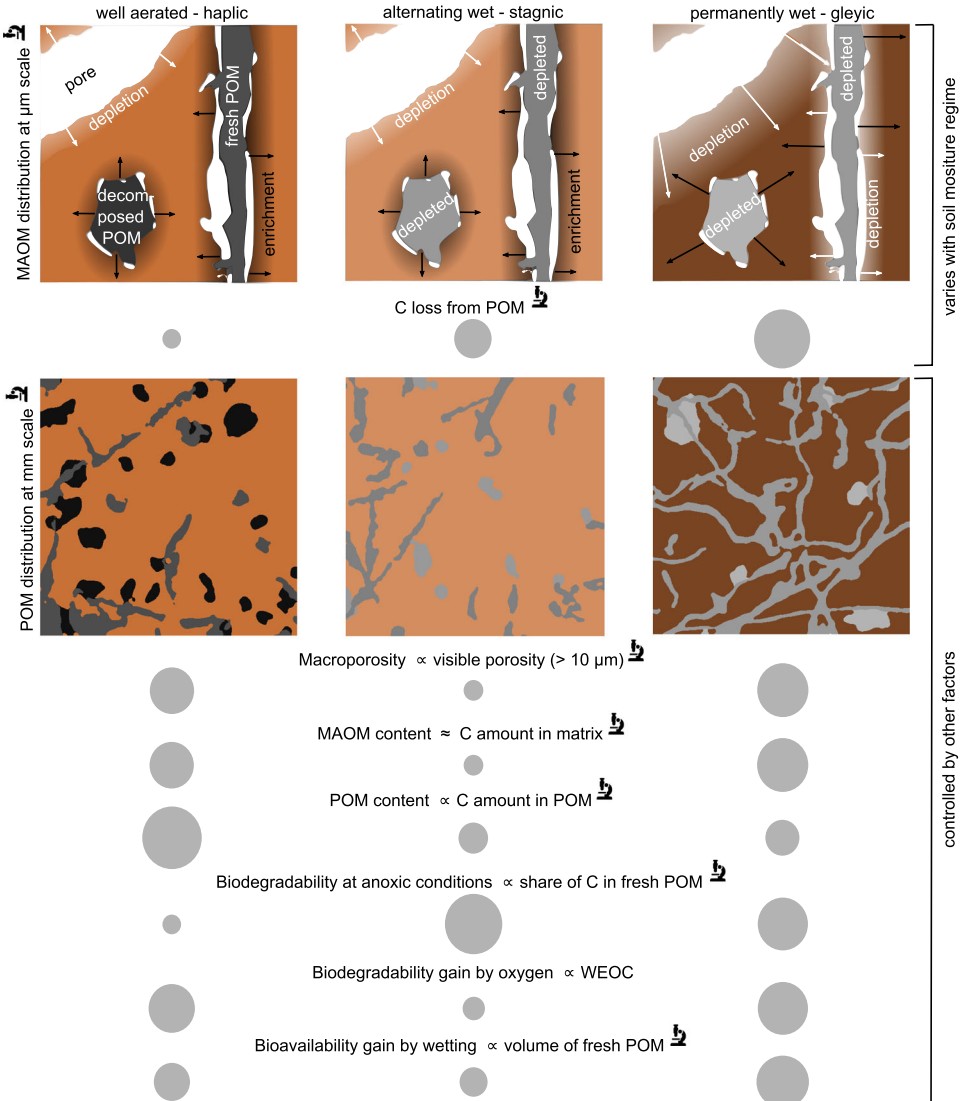

**Fig. 8 Synoptic overview on which properties varied with the soil moisture regime.** C loss from particulate organic matter (POM) and the redistribution of C in the soil matrix at the μm scale were mainly (but not exclusively) governed by soil moisture regime. The remaining properties (macroporosity, mineral-associated organic matter (MAOM) and POM content, POM distribution at the mm scale, bioavailability, and biodegradability) were influenced by other factors, among them mineralogy, land use, and productivity. The size of gray circles represents the magnitude of macroscale properties. Morphological POM types are interpreted towards the decomposition stage (fresh, decomposed). Different shades of brown and gray represent C contents in MAOM and POM, respectively. The microscopy icon, ∝, and ≈ represents image-derived properties, "proportional" and "approximately", respectively.

fresh POM as a major source of WEOC. The gain in biodegradation from anoxic to oxic conditions indicated how much more efficiently organic compounds can be mineralized along optimal metabolic pathways than along less favorable ones. This biodegradability was in good agreement with WEOC contents (Fig. 7c).

POM is not only a C input to soil, but also enhances oxygen supply due to incomplete contact with the surrounding soil and volume loss during decay. The large majority of POM particles were in direct contact with visible pores (aperture>10 μm, Fig. 6b). Seemingly, the internal porosity and small gaps around POM promote better aeration of POM than of the soil matrix. In addition, decomposed POM was more fragmented and occluded due to longer exposure to soil-structure turnover, than fresh POM residing in a well-connected network of elongated pores with low capillarity (Fig. 6a). This network is bound to fall dry under partially wet conditions. This loss in bioavailability was assessed by testing the mineralization under oxic, fully wet conditions (100% water-filled pore space (WFPS)) and oxic, partially wet (50% WFPS) conditions. The reduction scaled linearly with the volume fraction of fibrous POM and was most dominant in the Gleysol with the highest abundance of fresh litter and roots located in pores with low capillarity where hydraulic connection to the surrounding soil matrix is lost under dry conditions[23,26]. This was also reflected by the shortest distance of fibrous POM to air-filled pores at field capacity among the investigated soils (Supplementary Fig. 7). It was suggested recently that decomposing plant tissue may absorb water from the adjacent soil matrix[56,57] and thereby maintain high microbial activity even under partially wet conditions. However, the capacity for water absorption is reduced by water repellency, which in turn is modulated by the quantity and quality of organic matter[58]. More incubation experiments under well-controlled conditions with plant residues embedded in their intact structure and with more soil types than presented here are required to assess the water absorption effect.

In summary, the morphological approach of estimating the volume of (and C stored in) fibrous and compact POM can be interpreted towards decomposition stages as it yields accurate predictions of the bioavailability and biodegradability under various standardized conditions in short-term incubations in line with the previous studies[23–26]. In addition, it provides microscopic clues to long-term C turnover under different moisture regimes.

## Methods

**Soils characteristics**. Topsoil material from fine-textured soil was collected from three sites in Germany, either managed as grassland or agricultural soil for several decades. Plant compositions and crop rotations are summarized in Supplementary Table 1. A Stagnic Luvisol derived from Loess was sampled from long-term cropland at Rotthalmünster at the foothills of the Alps (360 m a.s.l., 856 mm, 8.2 °C). It was low in total organic C (12.0 mg C g$^{-1}$ soil) and contained mottling features induced by slow drainage. A Haplic Chernozem from Loessian deposits under dry, continental climate (116 m a.s.l., 484 mm, 8.8 °C) was sampled from long-term cropland at Bad Lauchstädt. It had a higher organic C content (22.1 mg C g$^{-1}$ soil) and a sizable amount of fossil C due to atmospheric deposition[59]. The Fluvic Gleysol on sandy clay loam was sampled from long-term grassland in a flood plain near Gießen (172 m a.s.l., 600 mm, 9.3 °C) and had by far the highest organic C content (39.5 mg C g$^{-1}$ soil), yet with the lowest content of occluded POM and the largest amount of WEOC in the mineral matrix (Supplementary Table 1). The soils were characterized with respect to texture, clay mineralogy, cation exchange capacity, extractable Fe and Al contents, organic matter fractions, soil respiration, and denitrification in associated experiments (Supplementary Table 1)[12,60,61]. In brief, bulk soils were fractionated according to density using sodium polytungstate solution adjusted to 1.6 g cm$^{-3}$ to separate POM (< 1.6 g cm$^{-3}$) from MAOM (> 1.6 g cm$^{-3}$). The MAOM fractions were analyzed for TOC using a Vario Max Cube; analyses of POM were carried out with a Vario EL cube (Elementar Analysensysteme GmbH, Langenselbold, Germany). Since density fractionation might release soluble C from plant residues, the electrostatic attraction was used[62] to remove loosely bound POM material. This POM material was ground to <1 mm for WEOC analysis of POM from bulk soil, which was determined by suspending 500 mg of POM in 25 ml ultrapure water and shaking for 1 h. After centrifugation at 3000 × g for 10 min (Cryofuge 8500i, Thermo Fisher Scientific, Waltham, MA, United States), the supernatants were passed through 0.45-μm membrane filters (Supor-450, Pall Cooperation, New York, NY, USA) and analyzed for dissolved organic C with a multi N/C 3100 instrument (Analytik Jena AG, Jena, Germany). Soils were dry-sieved and eight soil aggregates in the size fraction of 4−8 mm were collected from each soil for image analysis. The mean aggregate diameter was 6.5 mm (assuming a sphere with equivalent volume, min: 5.9 mm, max: 7.3 mm) across all soils. Additional soil aggregates were analyzed for TOC and total nitrogen (TN) using a Euro EA Elemental Analyzer (HEKAtech GmbH, Wegberg, Germany) with a nondispersive infrared sensor. Likewise, 5 g of soil aggregates were suspended in 25 ml ultrapure water and subjected to the same workflow as described for WEOC analysis of POM in order to determine WEOC of soil aggregates.

**X-ray computed tomography**. A total of 24 samples (eight samples per soil) were scanned with X-ray CT (X-tek XTH 225, Nikon Metrology, 80 kV, 75 μA, 1 s exposure time, no filter, 2400 projections, two frames per projection) and reconstructed in 16-bit at a voxel resolution of 5 μm with the X-tek CT pro software (Nikon Metrology). The smallest detectable pore diameter is in the range of 10–15 μm due to image processing. Scans of the same sample were acquired before and after Os staining to map the spatial Os distribution.

**Osmium staining**. The soil aggregates were placed between two glass Petri dishes together with solid OsO$_4$ (ReagentPlus 99.8%, Sigma Aldrich) at a concentration of 0.06 g OsO$_4$ g$^{-1}$ soil. The soil aggregates were exposed to the OsO$_4$ vapor for 2 days in a sealed desiccator under a fume hood, which was enough for the entire OsO$_4$ to vaporize and diffuse into the soil. Complete penetration of the aggregates with Os was later confirmed with X-ray CT. The total amount of C stained by Os varies between >1% for fresh root material and 2−3% for organic matter in soils[35].

In addition to the soil aggregates, eight reference materials were used to determine Os sorption to typical soil components: three solid materials (1. aluminum, 2. quartz sand (0.4−0.8 mm, Roth, Germany), 3. polyvinyl chloride fragments), three fine-grained minerals (1. silica fine flour (SIKRON SH 200, Quarzwerke, Germany), 2. illite (green shale, Ward's Science, USA), 3. goethite), and two porous organic materials (1. milled maize straw, 2. commercial hardwood char (Favorit, Alschu GmbH, Germany)). Goethite was prepared by raising the pH of 1 M FeCl$_3$ • 6 H$_2$O solution to pH 12, followed by the aging of the precipitates at 55 °C for 24–48 hours until the color had changed completely to ocher (the modified procedure by Atkinson, Posner[63]). The reference materials were filled into aluminum rings and exposed to 0.5 g OsO$_4$ (SERVA Electrophoresis GmbH, Germany) vapor for three weeks. X-ray CT imaging was done before and after Os staining at a resolution of 19 μm (130 kV, 150 μA, 0.708 s exposure time, 1 mm copper filter, 2000 projections, two frames per projection).

**Soil sections**. Four soil aggregates of each soil were impregnated with a series of Araldite 502: acetone mixtures (1:3, 1:1 (vol:vol)) and finally with 100% Araldite 502 (Araldite kit 502, electron microscope sciences, Hatfield, USA)[64,65] and the blocks were cured at 60 °C for 48 h. X-ray CT scans of the embedded samples were acquired after resin impregnation and prior to sectioning to check for internal deformations during Araldite application. The impregnated soils were cut with a low-speed saw (Struers Discoplan TS) in equidistant, parallel sections. The resulting 3–4 soil sections per aggregate were glued onto a quartz glass sample holder. The fixed aggregate slices were ground down and subsequently polished to obtain thin sections with a surface of low topography to allow for high-resolution imaging using μXRF and NanoSIMS. In order to determine the average thickness of soil sections, the discs were stacked and scanned jointly with X-ray CT at a spatial resolution of 15 μm.

**X-ray fluorescence microscopy**. A total of 36 soil sections (Luvisol: 11, Chernozem: 11, Gleysol: 14) were mapped with μXRF (Micro-XRF Spektrometer M4 TORNADO, Bruker) at 50 kV, 500 μA, 15–20 ms pixel$^{-1}$ map time, 20 μm spot size, 8 μm pixel size). The raw counts were stored in 16-bit and normalized to a map time of 100 ms per pixel. Maps of various elements (Al, Ba, Ca, Cl, Cr, Cu, Fe, K, Mg, Mn, Na, Ni, Os, P, Rb, Si, S, Sr, Ti, Zr) were extracted. The higher the atomic number, the larger the depth from which excited electrons still emit photons. This resulted in different average Os counts for soil sections with different depths. This was accounted for by normalizing Os counts in soil with average Os counts in reference areas placed in exterior resin. This is feasible because sample preparation for resin embedding after Os staining inevitably leads to some Os remobilization[32]. Note that this approach is more accurate than normalization by the section thickness measured with X-ray CT, since the relationship between thickness and cumulative photon emission from all depths of the section is nonlinear. The X-ray CT images obtained after resin impregnation were registered into the microscopy plane mapped with μXRF by employing a landmark-based registration protocol[66] for the elastix image registration software[67]. The Si map was used for registration as it correlated best with the structure information of the X-ray CT. Aluminum and Fe intensities were normalized by Si intensities (Al/(Al +Si) and Fe/(Fe+Si)) to a range of 0–1 and average normalized intensities were calculated as a function of Euclidean pore distance after pore segmentation of the registered X-ray CT image via Otsu thresholding[68] in Fiji/ImageJ[69] (more information on image registration given below).

**Secondary ion mass spectroscopy**. Two selected soil sections of aggregates from Stagnic Luvisol and Fluvic Gleysol were investigated using a NanoSIMS 50 L instrument (CAMECA, Gennevilliers, France). Briefly, the Cs$^+$ primary ion beam hitting the sample with an impact energy of 16 keV induces releasing of secondary ions from the surface. To avoid charging during the NanoSIMS analysis on nonconductive material, the samples were coated with an Au/Pd layer of ca. 30 nm (Sputter-Coater Polaron Emitech SC7640). Prior to NanoSIMS measurements, contaminants and the Au/Pd coating layer were locally sputtered away using a high primary beam current (pre-sputtering/implantation), while the reactive Cs$^+$ ions were implanted into the sample until the secondary ions reached an equilibrium steady state. In addition to the conductive coating Au/Pd layer, the electron flood gun of the NanoSIMS was used for charge compensation at the measurement time. The primary beam focused at a spot size of ~150 nm (ca. 2 pA) was scanned over the sample on areas of 30 μm × 30 μm, and $^{16}O^-$, $^{12}C^-_2$, $^{12}C^{14}N^-$, $^{27}Al^{16}O^-$, and $^{56}Fe^{16}O^-$ and $^{192}Os^-$ secondary ions escaping the sample surface were collected on electron multipliers with an electronic dead time fixed at 44 ns. The secondary ions $^{12}C^-_2$, $^{12}C^{14}N^-$ are a proxy for organic matter, $^{16}O^-$ and $^{27}Al^{16}O^-$ are indicative of mineral matrix, $^{56}Fe^{16}O^-$ reveal Fe-rich areas, and $^{192}Os^-$ trace the incorporated Os.

In all, 15 selected spots across both soil sections were measured in distinct microenvironments previously registered as nodules (n = 4), mineral matrix (n = 6), or plant residues (n = 5). Further parameters were: dwell time of 1 ms pixel$^{-1}$, 256 × 256 pixels for a 30 × 30 μm field of view with 40 planes per measurement. All planes were accumulated in a single plane after drift correction using the OpenMIMS plugin in Fiji/ImageJ. Absolute counts of a specific ion were normalized by the sum of all seven ion maps in each pixel[29] in Fiji/ImageJ in order to ensure comparability among the 15 selected spots. Co-localized ion intensities ($^{192}Os^-$ ~$^{12}C^{14}N^-$, $^{192}Os^-$ ~$^{27}Al^{16}O^-$, $^{192}Os^-$ ~$^{56}Fe^{16}O^-$) were evaluated after downscaling the pixel size to 0.6 μm in Fiji/ImageJ in order to reduce high-frequency noise. Co-localization in individual spots was quantified as the variability in Os intensity explained by the variability of the other ion intensity ($R^2$ of the ion pair).

**3D Image processing**. X-ray CT images of the same sample before and after Os staining as well as after resin impregnation was registered onto each other with elastix[67] employing a landmark assisted similarity transform[70]. The spatial distribution of Os was determined by subtracting the image acquired before Os staining from the image acquired after Os staining followed by a Median filter to remove noise in the difference image (Supplementary Fig. 1) with Fiji/ImageJ. Prior

to this subtraction, the grayscales were normalized by linear rescaling with a fixed gray value of 40 and 120 for the plastic sample holder and quartz grains, respectively. A small number of areas in each reference material were selected by the ROI Manager in Fiji/ImageJ. The resulting arbitrary unit for $Os_{CT}$ in the difference image would thus be 80, if the presence of Os would increase the gray value from that of plastic in the image prior to Os staining to that of quartz after Os staining.

Supervised classification of the normalized raw images into pores, POM, soil matrix, and dense areas including rocks and Fe-rich concretions was carried out with the ilastik software[71]. A parallel random forest classifier was used in multidimensional feature space that included the original gray values as well as gradient (1st derivative of gray values) and texture information (2nd derivative of gray values) after Gaussian smoothing with a strength of $\sigma = [0.3, 0.7, 1.0]$. In this way, characteristic traits of each material like the aperture of cracks, the inherent heterogeneity of the organic fabric, or the homogeneity of quartz grains were harnessed for material detection. The classifier was trained with a few test lines for each material class. The outcome of image classification was denoised with a majority filter implemented in the QuantIm image processing library[72]. The aggregate boundaries were determined with the Adaptive rectangle tool on the grayscale data in VG Studio Max 3.4 (Visual Graphics) in order to impose the exterior as an additional material onto the segmented images. The POM material class was labeled with 'Connected Components Labeling' in the MorphoLibJ plugin[73] for Fiji/ImageJ. POM clusters $>6.25 \times 10^5 \, \mu m^3$ (5000 voxels) were analyzed with respect to different morphological traits using Analyze Regions 3D in MorpholibJ and stored in data tables. From these traits four different properties were derived to distinguish decomposed from fresh POM: elongation, compactness, plateness, and mean Os intensity (Supplementary Table 2). These morphological properties reflect the increasing fragmentation and shape evolution of POM from non-decomposed plant residues characterized by conserved cells structure to sub-rounded nodules of ~40 μm that is gradually mixed with the soil matrix as previously reported based on soil micromorphology[30]. Out of 5149 POM objects, 463 were manually assigned to fibrous or compact POM after visual inspection. The remaining objects were assigned with a random forest classifier in R[74] as follows: The pre-assigned objects were split into a training ($n = 324$) and validation ($n = 139$) data set. A classifier was trained on the training data set with 500 trees and two variables were tried at each split. Employing the classifier on the validation data set led to a prediction accuracy of 92%. Then a new classifier was trained on all available data ($n = 463$) resulting in an OOB (Out-of-bag) estimate of an error rate of 8%. This classifier was applied to the entire data set. The final assignment was imposed on the label image via the 'Assign Measure to Label' method in MorpholibJ. The Os amount bound to matrix-bound organic matter and POM was calculated as the product of average Os intensity in and the volume fraction of this material class. Likewise, the share of C in fibrous POM was calculated as the ratio between Os amount in fibrous POM and the sum of Os amounts in fibrous and compact POM. Euclidean distance maps were calculated on binary images in Fiji. These represent the Euclidean distance to the closest foreground voxel, e.g. pores or POM, in all background voxels. Finally, Os intensity in the difference image, $Os_{CT}$, was averaged for different material classes as well as for different distances in certain material classes. The spatial extent of Os depletion or enrichment in the soil matrix was determined as the Euclidean distance at which the Os intensity reaches the arithmetic mean ±1 standard deviation of the bulk soil.

**Short-term soil incubation**. For oxic incubations, 3.5 g of air-dried soil aggregates (diameter 2−8 mm, 15−20 aggregates per bottle) were loosely placed on the bottom of 100-ml injection bottles (Chroma Globe, Kreuzau, Germany) to assure that gaseous oxygen can reach aggregates without constraints induced by touching aggregates. The dry aggregates were carefully rewetted to 50% WFPS by dripping water with a syringe. After pre-incubation for seven days under aerobic conditions at 20 °C to stimulate microbial growth, soil aggregates were brought to different saturations (50% WFPS—partially wet, 100% WFPS—fully saturated) with a syringe. Subsequently, incubation bottles were sealed with a chlorobutyl rubber stopper and crimped with an aluminum cap. After evacuating (<250 mbar) and flushing the bottles three times with synthetic air (Air Liquide, Düsseldorf, Germany), soil aggregates were incubated at 25 °C in the dark. Gas samples were taken after 0, 24, and 72 h, and analyzed for $CO_2$ with gas chromatography (Agilent HP 7890B). Gas emissions under anoxic conditions were carried out in a slightly different spatial setup to concur with a precursor study. Here, macroaggregates (2−8 mm) were embedded in quartz silt (<125 μm) and rewetted to 40% WFPS with ultrapure water and pre-incubated for seven days under aerobic conditions at 25 °C in the dark. Subsequently, WFPS was set to 60%, which is considered optimal for substrate availability in the aggregates and gas diffusion in the silt matrix. The actual water saturation in aggregates might be higher in aggregates due to the stronger capillarity induced by the clay fraction. Then, the macroaggregates were incubated anoxically at 25 °C in the dark. Preparation of anoxic atmosphere in the incubation bottles, gas sampling, and gas analysis are described in[75]. Rates are reported for the period from 24 to 72 h. All incubations were carried out in triplicate. The spatial proximity of soil aggregates and gas diffusion through the quartz silt matrix is assumed to be irrelevant for respiration under anoxic conditions. If the presence of the silt matrix would have had an effect on anaerobic respiration, it had been the same for all three soils and therefore without an effect on the relative differences in respiration ratios between soils.

**Statistical analysis**. One-way analysis of variance (ANOVA) in combination with Tukey's Post-hoc tests was carried out in R[76] in order to test for significant differences in $Os_{CT}$ intensities between soils. For comparisons of $Os_{CT}$ intensities in POM particles and pore distances of POM particles, two-way ANOVA was carried out with POM type as the second factor in addition to soil type. Comparisons were done at the level of individual aggregates ($n = 8$ per soil type), i.e. after averaging across all POM particles per soil aggregate. Significance was tested at $p < 0.05$ unless reported otherwise (***$p < 0.001$, **$p < 0.01$, *$p < 0.05$, ·$p < 0.1$, n.s. $p > 0.1$). The normality of the residuals was checked with Shapiro-Welch tests and equality of variances with Levene's test. If necessary, values were log-transformed in order to ensure the normality of residuals.

## Data availability
The source data underlying Figs. 2–7, as well as Supplementary Figs. 6–9 generated in this study are provided in the Source Data file. Source data are provided in this paper.

## Code availability
Segmented X-ray CT data will be uploaded to the Soil-Structure Library hosted by the UFZ (https://structurelib.ufz.de/). Additional image data, as well as image analysis codes, are available from the authors upon request.

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

## Acknowledgements
This research was funded by the German Research Foundation through two research grants (416883305, 290269257). We thank Claudia Stehr at the Fraunhofer Institute for

Microstructure of Materials and Systems IMWS in Halle for Osmium staining and the Höhere Landbauschule Rotthalmünster for access to field sites.

## Author contributions

This work was originally conceived by S.S. and H.V. with additional input from C.W.M., R.M., R.S., and K.K. Laboratory experiments were conducted by S.S., F.L., and R.S. Imaging was carried out by S.S., C.H., R.K., and C.W.M, and image data were analyzed by S.S., F.L., L.A., C.H. The manuscript and supporting information were written by S.S. with input from all coauthors.

## Funding

## Competing interests

The authors declare no competing interests.
