## [Peer Review File · Nature Communications]

REVIEWER COMMENTS

Reviewer #1 (Remarks to the Author):

The manuscript addresses an extremely important topic of micro-scale mechanisms behind soil C storage and protection. Until recently this topic has received very little attention, resulting in poor understanding of what drives soil C gains and how to maximize them for curbing climate change impacts. The reason for lack of attention to micro-scale soil processes are substantial technical challenges in exploring physical and chemical characteristics of intact soils at micro-scales. This study combined state-of-the-art techniques to successfully address several of such challenges and to gain important insights into the role of soil pore architecture in defining spatial distribution patterns of soil C. I would like to especially compliment the authors on a very interesting and important work they did in modifying and exploring the Os staining approach for soil organic matter mapping. I believe this work will be of great interest to a very wide audience of environmental scientists. The study appears to be very thoroughly done and the manuscript is well written.

I strongly recommend it for publication, after a revision. My specific suggestions for improvement are listed below.

Abstract I.18 - While I understand the word limit constrains, still I believe it will be beneficial to briefly mention the essence of the novel protocol and the specific techniques used in the study in the abstract.

I.22 - I do not believe that it is correct to attribute the differences the authors observed among the three studied soil types solely to the differences in the soil moisture regimes. If understand correctly, two of the studied soils are from long-term cropping, subjected on a regular basis to tillage disturbance (as per I.268). While the third soil is from a long-term grassland. This drastic difference in land use history is strongly reflected in the reported soil organic C levels, and I am sure it had a very substantial influence on how pores are formed, where plants deposit new C, how easily mineralized that C might be. The differences in management likely have had as large or even larger effect than the differences in soil moisture regimes. The fascinating differences in the spatial patterns of Os stained organic as a function of distances from POM (Fig. 5) to me can be well attributed to cropped vs. intact grassland land use effects.

I.64 I believe it is a bit of a stretch to call POMs of these two different shape types "fresh" and "aged". A lot of fresh POM of non-root origin might have rounded shapes - e.g., animal remains, aboveground plant residues that got belowground, root nodules, etc. I believe it will be more accurate to refer to those two types of POM as "root-origin" and "other-origin", adding that the other-origin group can also include old substantially decomposed roots, biochar, and be in general of greater age.

I.71-72 Do you have evidence/citations to support this concept? I believe OM losses because of exchange with soil solution might be rather minor. I would think that the differences in microbial activities, which you discussed earlier, within the pores or close to pores is what might be playing a much bigger role here.

I.102-104 To me this is one of the most important outcomes of this study. Thus, I would like to see it not in supplemental but in the main paper and presented with much more detail. I am particularly interested in the apparently high Os absorption by clay minerals. This is not something that has been reported in synchrotron dual energy scanning studies of Os staining before. Potential effect of such absorption on Os staining in soil needs to be discussed, since it does influence what can be inferred from the spatial patterns in Os distributions.

I.108-110. It would be nice to also see the regressions between Os and NanoSIMS C (from images on Fig. 3).

I.150-151. But would not Os concentrations also vary due to Fe and Al-based minerals as you described above?

I.152-153. Simplify the sentence - e.g., "a negative linear relationship between x and y"

I.155-157. Fascinating! Could it be because of the differences in plant species recently grown in the studied sites? I suggest providing information on the dominant plant species in the past few years.

I.172-173. I am not sure I fully understand here. You mean there are fewer Fe and Al minerals in vicinity of pore boundaries on which Os also can sorb? Because if all mineral surfaces are taken by the absorbed organic, then the Os would be attaching to that organic, no? Or are you saying that because there are fewer Fe and Al mineral surfaces then there is less MAOM?

I.235-236. I presume that you do not have the soil C measurements from individual aggregates? That is unfortunate, because, given the very large range of observed total C values in these three soils (i.e., the highest / the lowest^{~4}), it is not surprising that all C related variables are strongly correlated to each other. But if you do have the individual aggregate data, it would be informative to examine these relationships within each soil type.

I.257 - Can this statement be supported by references?

I.278-279. Are there data/references to support this statement? Also, besides potential differences in total C contents, what might be more important is that the aged POM contains more of recalcitrant organic compounds than the fresh plant residues.

I.280-281. But you are not seeing differences in Os-absorption between fresh and aged, are you? To me, lack of such consistent differences in Os staining is one of the reasons to be more careful with this fresh vs. aged POM classification.

I.290-292. Not sure I follow why there would be a contradiction. POM C is mineralized when more oxygen is available in biopores, but then the decomposition products diffuse away from the POM (as you show) and become protected at some distances from POM within the soil matrix, no?

I.311. While I agree that there is no need to repeat in detail the method descriptions that were reported

elsewhere, still I believe it is much needed to at least state which methods were used, and possibly provide key method details, for the soil variables that are part of the Results and Discussion in this specific manuscript. Different fractions of POM? WEOC? How all these were measured?

In addition, here are some other things that I believe would be very beneficial to add to the Methods, and to potentially consider when discussing the results: 1) Depth from which the soil aggregates for the study were collected. 2) The management details, such as tillage and recent crops. 3) Length of time under current land use.

Since the study is focusing on soil moisture regimes as the potential driving force for the observed differences, it would be good to provide more information and show some data describing the differences among the regimes.

I.687-688. Maybe separate the portion with POM examples as, say, part d of the figure?

Fig. 2. Methods used for TOC, MAOM, POM, and WEOC measurements need to be listed in the Methods. Also please describe how the share of C in fresh POM was determined.

L.702 Is this what pink, blue, green colors represent? Please add a description.

I.706. Fig. 4. Can the matrix/POM/dense portions on part b be directly compared to each other? It is very surprising that in Gleysol the POM has lower Os than the matrix.

I.711-712 and the Statistical analyses in general (I.453-459). I find it very surprising that such seemingly small differences among the means with such high variability turn out to be all statistically significant. Are you sure that the correct error terms were used in the stats and that the individual POM pieces were not treated as experimental units in this analysis? Please add a description of how the hierarchical nature of the study's experimental design (individual POM pieces, aggregates, soil types) was accommodated in the statistical models used for the data analyses.

Fig. 6a - I recommend adding more separating lines between different WFPS and anoxic and oxic conditions - to make the figure easier to read. Adding an x-axis line also will be helpful.

Fig. 6b. Not sure I fully understand the measure shown on the y-axis. Is this a ratio of how much C was emitted during anoxic incubation in 1 day divided by the total soil C? Does this mean that in some cases 60% of TOC respired in 1 day?

Reviewer #2 (Remarks to the Author):

This study examines the microscale distribution of particulate and mineral associated organic matter

(POM and MAOM) in soil types with different drainage regimes. An Osmium (Os) staining approach is used for the first time in combination with a conventional (non-synchrotron) X-ray CT scanner to map the distribution of organic matter within aggregates from each soil. Additional NanoSIMS analyses as well as lab incubations under oxic and anoxic conditions and at different moisture contents are performed to corroborate some of the findings. The authors nicely demonstrate the ability of Os staining and CT imaging to discern fresh and aged POM, with contents varying across the soil types. Os staining within the remaining soil matrix (= everything not considered POM) was defined as MAOM. Os staining also revealed gradients of organic matter around POM fragments as well as pores. Generally, organic matter contents within the soil matrix (as indicated by Os concentration) increased towards POM and decreased around empty pores. The NanoSIMS results seem to indicate that Os stains not just organic matter, but also Fe oxides and clay minerals, which complicates the interpretation of Os as a proxy for organic matter. The incubation results showed a relationship between degree of POM and POM-derived leachate and microbial respiration under anoxic conditions.

This work is an extremely thorough and multifaceted investigation of the 3D distribution of organic matter in soils. How the physical architecture of soils influences soil carbon storage is one of the key research questions in soil science. By combining a unique suite of techniques, this study offers some interesting and novel insights. For example, the fact that visualization and quantification of organic matter distribution around POM and empty pores. The connection between POM leaching and anoxic CO₂ production is also noteworthy.

While I believe this manuscript would be of interest to the broader soil science community, the following points could be clarified/improved upon:

1) The classification of everything that is not POM as MAOM. Given the recent attention the simple, functional classification of organic matter in POM and MAOM pools has received, I understand the desire to partition POM and MAOM in this study. I applaud the authors on this attempt and think it is a useful first step. But I have a hard time with the idea that everything that is not POM in the CT scans can be classified as MAOM. First of all, the 5 μm voxel size is just too large to truly resolve MAOM, which we would expect to be associated with minerals such as Fe oxides and clays, which are smaller than 2 μm . Second, if Os binds to exactly those minerals, how do we know Os staining signifies MAOM, and not just the presence of these minerals in a certain soil locale? Perhaps it would be more appropriate to present this type of OM as "matrix-bound OM", which could be truly mineral-associated, but also free amorphous OM, POM smaller than the spatial resolution, or microbial biomass.

2) The assumption that soil moisture is the main driver of the differences in POM/MAOM patterns across the three soils. The three soils don't just differ in the moisture regime, they will also have vastly different C contents, MAOM/POM fraction, C inputs, degrees of bioturbation, mineral composition, etc - all of which will alter the MAOM/POM fraction. I think it is misleading to assume that the observed differentiation in MAOM/POM distribution are solely driven by differences in moisture dynamics. It is not like soil moisture is manipulated, and all other factors remain constant. I don't argue it doesn't play a role, merely that soil structure and the emerging distribution of MAOM/POM within these soils will be a function of the interaction between soil moisture AND these other variables (C inputs, bioturbation, mineralogy, etc.). I think it would help to state more clearly from the beginning that this study compares

soils with significantly different characteristics that should result in differences in the spatial arrangement of the organic matter within the pore system.

3) Make it clear from the onset why NanoSIMS and incubations were used. Reading through the manuscript, it came as a surprise to see that NanoSIMS, XRF, and additional soil incubations were used, and I wasn't always sure how they were related to the original CT measurements. Perhaps it could be useful to justify in the introduction why this multifaceted approach was used.

4) Incubations not conducted under comparable conditions. I'm a bit concerned about the ability to compare the oxic and anoxic incubations if they weren't conducted under the same conditions. Some soil aggregates were embedded, others weren't, but it's unclear why, and it doesn't give me confidence that the experiment really succeeds in isolating O₂ availability as a factor.

5) I had a hard time following the manuscript. There are a lot of different data types and the presentation often wasn't very clear. What I found particularly confusing is the fact that results and discussion aren't fully separated. The many different results are presented bit by bit, but then discussed individually. In the actual discussion, these many different pieces aren't really brought together and synthesized. So, when the conceptual model is presented, it's not very clear where that is coming from. I think it would help if the results were more compact and to the point, and were then followed by a discussion that (i) synthesizes the results and (ii) articulates how these results informed the conceptual model that was developed.

Given those limitations, I wonder if there is a story the authors could develop around the visualization of POM, its decomposition and penetration of the surrounding soil matrix, and its role in mediating C mineralization under varying O₂/moisture contents. That would be a very interesting story (to me) that could be told with or without explicitly mentioning MAOM.

Abstract

19: What is the "thickness of MAOM depletion"? Can depletion have a thickness?

Also, depletion relative to what? Are you saying the layers become less thick?

Finally, what do you mean by "around pores"? Any pores as measured by X-ray CT, micropores, macropores?

Intro

29-34: If that's already known, what is really the knowledge gap explored here?

Hypothesis seems a bit like a strawman – is it truly falsifiable?

56-59: Seems like the actual insight gained from this study. Maybe emphasize more?

66-72: These soils will differ in mineralogy, structure, C inputs, etc – IE more than one factor (soil moisture regime) varies, which makes it difficult to attribute the observations solely to variations in soil moisture. It would be more appropriate to state clearly (e.g. in the abstract) that soils across a drainage gradient were examined and showed the pattern reported here.

Fig. 1: The text talks a fair amount of MAOM identified in Fig. 1, but I can't see anything labeled as MAOM here. Please explain. Also, how do you know the nodules are Fe-rich?

100: abbreviate as POM?

105-107: X-ray CT quantification of POM seems good (significant), but that of MAOM is not great (not significant) (Fig. 2). Makes sense because it's just quantified by subtraction. Maybe only highlight POM, which can be quantified with higher confidence?

119: "perfectly"? Maybe a bit much with three data points.

127-129: Isn't the fact that Os binds to exactly those minerals that are principally responsible for MAOM formation (Fe oxides and clay minerals) a problem? How do you know if it binds because of the affinity for the mineral as opposed to the presence of MAOM?

132-138: Ok, so I understand you're trying to use NanoSIMS to resolve that. But NanoSIMS just seems to reinforce that point, i.e., Os binding to clay minerals and Fe oxides? How do you know it's MAOM?

143: Point well taken

147: Why can relative differences among soils still be interpreted? What if one soil has more Fe oxides/clays than the other? That would lead to larger "MAOM" estimates even if there is not OM bound to the minerals due to the affinity of Os to the minerals. I'm sure you've thought this through, but I'm not following ...

176: it would be useful to know on what grounds that possibility was ruled out

Fig. S5: It would help to see how well the signal intensity is correlated. The maps don't seem that similar and there are hotspots in both maps that only show in one or the other.

177ff: isn't it possible that this is an artifact of the X-ray CT technique and where the threshold is drawn for the segmentation? The boundary between "pore" and "matrix" might not always be super clean, right? And, consequently, the density of the material surrounding the pores might be less dense and thus bind less Os? I don't know much about X-ray CT, but the fact that this "depletion" occurs right at the segmentation threshold that is always tricky to get right raises some questions for me.

189 and 192: these statements are examples where I think focusing on the results without further

discussion would be helpful. These statements clearly belong in the discussion and, in the present form, don't feel very well supported and appear speculative.

Fig. 5a. I suggest labeling the box plots (b or c) as well so you can refer to them in the text more easily. Otherwise, it remains difficult to assess which aspect of 5a is discussed in the text.

205: Is it possible that Fe and Al are less abundant in the vicinity of POM in the gleysol and that's why you are observing less Os binding? It would seem that POM would be a hotspots for leaching of both Al and Fe due to pH or redox driven leaching of both.

Section 2.3 header: I would suggest making it clear this is a results section. "implications..." made me think this is part of the discussion.

221: p value is different from what is shown in Fig. 6b. Also 6b only shows fresh POM, so this sentence is a bit confusing.

Discussion

I believe that this paper would really benefit from a concise summary of the results before the presentation of the emerging conceptual model. There are a lot of results to digest, and it would be helpful to the reader to know how these data come together to paint the picture you are presenting in the following.

240-252: This conceptualization disregards the role that differences in the mineralogical composition of the soils might play in the formation of MAOM in these soils. This overview also doesn't take into account the incubation results, which reinforces the disconnect between the imaging results and the incubation.

284: This is the conclusion/summary paragraph. Yet, I believe this is the first time biopores are mentioned. Another complicated concept that needs more introduction or should be omitted to avoid further complicating the story.

437: WHC reported here, but WFPS reported in Fig. 6. Which one is it?

445: How does 40 or 60% WFPS compare to the WHC numbers reported above?

446: On what basis? The quartz silt mixture is going to hold some water and influence gas transport, so alter the dynamics in the soil aggregates. I don't understand why the oxic and anoxic incubations weren't performed under the same conditions if they are to be compared. Some justification or validation of the assumption that these treatments can be compared is necessary.

We thank the editor and the two reviewers for their valuable comments that helped to improve the quality of the paper. In the following we respond to all comments and describe the actions taken. Comments are shown in normal font and responses in bold, italic font. The line numbers with respect to actions taken refer to the document with tracked changes.

EDITOR COMMENTS

As you will see from the reports copied below, the reviewers raise important concerns. Without thorough revisions to address these points, we are unlikely to send the manuscript back to review. In particular, there were concerns about the identification of soil moisture as the main driver of the patterns observed here, and other factors, such as land management history, should be considered and discussed. The comparability of the oxic and anoxic incubations also needs to be clarified, and the results should be appropriately caveated. Please also ensure all statistical analyses presented here are robust and well-supported (e.g. Reviewer 1).

We thank the editor and the two anonymous reviewers for their valuable comments that helped to improve the quality of the paper. We have now given a full appraisal of the interplay between soil moisture regime and land use. We have also carved out more carefully which experimental observations can be explained by soil moisture regime (carbon loss from particulate organic matter (POM), microscale carbon distribution around pores and POM), and which are controlled by other factors such as land use, parent material, and crop productivity. We also clarified the effect of different soil aggregate arrangements between oxic and anoxic incubations. In addition, we have corrected the statistical analyses and explained them better.

Moreover, we made a few adjustments to meet the formatting instructions of the journal, such as number of words in the title and abstract, figure styles and formats, etc.

REVIEWER COMMENTS

Reviewer #1 (Remarks to the Author):

The manuscript addresses an extremely important topic of micro-scale mechanisms behind soil C storage and protection. Until recently this topic has received very little attention, resulting in poor understanding of what drives soil C gains and how to maximize them for curbing climate change impacts. The reason for lack of attention to micro-scale soil processes are substantial technical challenges in exploring physical and chemical characteristics of intact soils at micro-scales. This study combined state-of-the-art techniques to successfully address several of such challenges and to gain important insights into the role of soil pore architecture in defining spatial distribution patterns of soil C. I would like to especially compliment the authors on a very interesting and important work they did in modifying and exploring the Os staining approach for soil organic matter mapping. I believe this work will be of great interest to a very wide audience of environmental scientists. The study appears to be very thoroughly done and the manuscript is well written.

We would like to extend our gratitude for this positive appraisal of our work.

I strongly recommend it for publication, after a revision. My specific suggestions for improvement are listed below.

Abstract I.18 - While I understand the word limit constrains, still I believe it will be beneficial to briefly mention the essence of the novel protocol and the specific techniques used in the study in the abstract.

Agreed. In the revised version, we have now mentioned the combination of osmium (Os) staining, X-ray computed tomography and machine learning in the abstract (line 22-23) and made some cuts elsewhere so that we can keep the word count of 150 words.

I.22 - I do not believe that it is correct to attribute the differences the authors observed among the three studied soil types solely to the differences in the soil moisture regimes. If understand correctly, two of the studied soils are from long-term cropping, subjected on a regular basis to tillage disturbance (as per I.268). While the third soil is from a long-term grassland. This drastic difference in land use history is strongly reflected in the reported soil organic C levels, and I am sure it had a very substantial influence on how pores are formed, where plants deposit new C, how easily mineralized that C might be. The differences in management likely have had as large or even larger effect than the differences in soil moisture regimes. The fascinating differences in the spatial patterns of Os stained organic as a function of distances from POM (Fig. 5) to me can be well attributed to cropped vs. intact grassland land use effects.

Agreed. Reviewer #2 also pointed out to confounding effects with mineralogy, land use, etc. We addressed this issue in two ways: 1) We completely revised the discussion and conceptual figure in order to identify what we speculate to be mainly (but not exclusively) governed by soil moisture regime (C losses from POM, microscale gradients in matrix bound organic matter) and what is influenced by other drivers, such as mineralogy, land use etc. (MAOM content, POM content, POM distribution at the mm scale, bioavailability, biodegradability) (lines 413-419, Fig 8). 2) We now added a comment how land use could in fact also affect the C enrichment patterns around POM and conclude that disentangling land use from soil moisture regime would require a full factorial design (lines 390-398). A thorough comparison of microscale C distribution patterns in cropland and grassland of the Luvisol site is currently underway, but not yet completed.

I.64 I believe it is a bit of a stretch to call POMs of these two different shape types "fresh" and "aged". A lot of fresh POM of non-root origin might have rounded shapes - e.g., animal remains, aboveground plant residues that got belowground, root nodules, etc. I believe it will be more accurate to refer to those two types of POM as "root-origin" and "other-origin", adding that the other-origin group can also include old substantially decomposed roots, biochar, and be in general of greater age.

Although we admit the difficulties in assigning all sorts of POM into two very simplistic classes, we do not fully agree with the suggested relabeling. There are also inconsistencies with the suggested notions: 1. With the current morphological approach "root-origin" would also include barely decomposed aboveground litter mixed in by plowing. Such litter particles appear fibrous as well and not compact and rounded. To account properly for both input pathways, this class should be denominated as "plant-origin". This, however, applies to almost all POM, either fresh (undecomposed), decomposed or strongly transformed, e.g. by charring, or fragments incrustated by minerals. 2. As correctly pointed out by the reviewer "other-origin" also included decomposed roots and therefore hardly serve as proper discriminator. It is thus the degree of decomposition that sets the two classes apart. We see the merit of relabeling the classes more precisely, but not as suggested. Instead, we think "fresh" vs. "decomposed" are more appropriate class labels. Please note, we adopted this classification from a combined micromorphology + X-ray CT study (Elyeznasni et al.,

2012) that also used the decomposition stage as classification scheme. Their classes were 1) non-decomposed vegetal remains, 2) organic macro fragments and 3) organo-mineral nodules assemblages. We think that using a more simplified classifying scheme is well justified both in terms of number and labeling of categories.

I.71-72 Do you have evidence/citations to support this concept? I believe OM losses because of exchange with soil solution might be rather minor. I would think that the differences in microbial activities, which you discussed earlier, within the pores or close to pores is what might be playing a much bigger role here.

According to our understanding of processes both mechanisms would result in same patterns and it is difficult to quantify the different mechanisms involved under natural conditions. Still, we think we should not single out only one possible explanation in the abstract or introduction, but discuss both options in the Discussion section. In general, it is a well-established concept that mottling patterns in hydromorphic soils are an imprint of iron and manganese redistribution patterns brought about by the pore structure, with direct consequences for C desorption and resorption. Some citations demonstrating the role of pore structure on soil mottling were already given in the discussion (Schulz et al., 2016; Fimmen et al., 2008).

We now also add a citation for the experimental evidence that Redox-driven manganese and especially iron mobilization and translocation in soil links closely to the remobilization and translocation of organic matter (Hagedorn et al., 2000). We also add that the concomitant increase in pH under reduced conditions promotes C desorption from Fe minerals and clay minerals (Grybos et al., 2009) (lines 356-358). However, tracking such C redistribution at pore scale has been impossible so far, due to technical challenges in C mapping at the microscale.

I.102-104 To me this is one of the most important outcomes of this study. Thus, I would like to see it not in supplemental but in the main paper and presented with much more detail. I am particularly interested in the apparently high Os absorption by clay minerals. This is not something that has been reported in synchrotron dual energy scanning studies of Os staining before. Potential effect of such absorption on Os staining in soil needs to be discussed, since it does influence what can be inferred from the spatial patterns in Os distributions.

Agreed. The figure has been moved to the main paper and hydrogen bonds are discussed as a potential mechanism for Os sorption to mineral surfaces (line 289-290).

I.108-110. It would be nice to also see the regressions between Os and NanoSIMS C (from images on Fig. 3).

Agreed. We have now carried out regressions between relevant ions for all available NanoSIMS spots ($^{192}\text{Os} \sim ^{12}\text{C}^{14}\text{N}$, $^{192}\text{Os} \sim ^{27}\text{Al}$, $^{192}\text{Os} \sim ^{56}\text{Fe}$) and computed the R^2 separately for the three different domains (nodules, soil matrix, plant residues). We show that Al is a better explanatory variable for Os in nodules, whereas CN performs better for plant residues. The R^2 is similar in the soil matrix (line 163-166). These findings are summarized in a new supplementary figure (Figure S6).

I.150-151. But would not Os concentrations also vary due to Fe and Al-based minerals as you described above?

Agreed. We have revised this part completely. We now state that Os intensities in the soil matrix should not be interpreted as MAOM contents, when comparisons are made across soils with different mineralogy (line 290-294). In addition, we discuss whether the spatial distribution of Fe and Al-based minerals is random with respect to pore distances and POM distance (line 296-305).

I.152-153. Simplify the sentence - e.g., "a negative linear relationship between x and y"

Done.

I.155-157. Fascinating! Could it be because of the differences in plant species recently grown in the studied sites? I suggest providing information on the dominant plant species in the past few years.

Both crop rotations of the Luvisol site and Chernozem site are very similar (a combination of corn and cereals). This information is now added to Supplementary Table 1. We think that different degree of POM leaching due to very different soil moisture regimes is the most logical explanation here.

I.172-173. I am not sure I fully understand here. You mean there are fewer Fe and Al minerals in vicinity of pore boundaries on which Os also can sorb? Because if all mineral surfaces are taken by the absorbed organic, then the Os would be attaching to that organic, no? Or are you saying that because there are fewer Fe and Al mineral surfaces then there is less MAOM?

We agree that this should have been elaborated more precisely. We discuss two scenarios: 1) There are less Fe and Al minerals and consequently also less MAOM to all of which Os sorbs proportionally. This is falsified most prominently by the μ XRF results for the Chernozem (Supplementary Figure 8) 2) Fe and Al minerals have homogenous concentration with respect to pores distances, but MAOM concentrations are lower because of preferential mineralization or desorption along pore boundaries and the Os traces this pattern. We have now rephrased the two scenarios accordingly. This part is moved to the discussion section now (line 351-361).

I.235-236. I presume that you do not have the soil C measurements from individual aggregates? That is unfortunate, because, given the very large range of observed total C values in these three soils (i.e., the highest / the lowest^{~4}), it is not surprising that all C related variables are strongly correlated to each other. But if you do have the individual aggregate data, it would be informative to examine these relationships within each soil type.

Correct. Incubations and imaging were carried out on parallel samples. Please note that CO₂ efflux represents the cumulative emission from 15-20 individual soil aggregates within each incubation vessel. Retrieving volumetric POM contents for all aggregates would have resulted in excessive CT scanning.

I.257 - Can this statement be supported by references?

Frequent exchange of solution means leaching of reaction products, which promotes dissolution and desorption processes. We think this is basic chemical understanding of mobilization and transport of ions and compounds in soil and, e.g., the underlying concept of leaching and weathering processes. We have now added a statement with citations that the concomitant pH increase under reduced

conditions may promote C desorption in hydromorphic soils (see response to your comment “lines 71-72”).

I.278-279. Are there data/references to support this statement? Also, besides potential differences in total C contents, what might be more important is that the aged POM contains more of recalcitrant organic compounds than the fresh plant residues.

Regarding occlusion: Yes, the different levels of occlusion are directly supported by the horizontal boxplots in Figure 6. Regarding leaching: the lower Os adsorption to aged POM compared to fresh POM in the hydromorphic soils (but not haplic Chernozem) suggests that leaching of POM is a relevant process in hydromorphic soils (see comment below). It is true that the propensity for leaching is linked to the recalcitrance of organic compounds. We have now stated that the large fraction of biochar in the Chernozem is recalcitrant (line 341-344).

I.280-281. But you are not seeing differences in Os-absorption between fresh and aged, are you? To me, lack of such consistent differences in Os staining is one of the reasons to be more careful with this fresh vs. aged POM classification.

Thank you for the sharp observation. We agree, we should have interpreted the differences in Os adsorption between fresh and decomposed POM straight away in the Results section. In accordance with your other comment (“lines 711-712”, see response below) we have now computed average Os intensities of both POM types for individual aggregates (n=8 per soil, instead of hundreds of individual POM particles) and carried out two-way ANOVA (soil type x POM type) with these aggregate averages (Figure 6a). Here we see a clear interaction effect ($p < 0.01$), i.e. fresh POM being more stained compared to decomposed POM in hydromorphic soils and less stained compared to decomposed POM in the Chernozem soil. This is now shown directly in Figure 6a (right) and interpreted in the discussion section (line 338-344).

I.290-292. Not sure I follow why there would be a contradiction. POM C is mineralized when more oxygen is available in biopores, but then the decomposition products diffuse away from the POM (as you show) and become protected at some distances from POM within the soil matrix, no?

The contradiction is linked to the net C balance, i.e. whether or not there is a net C depletion or accumulation in the soil matrix around pores with a diameter of 30-150 μm . In other words, whether the improved C mineralization or C desorption around these pores exceeds the higher C input from POM and subsequent protection in the adjacent soil matrix. Kravchenko et al. (2019) have posited that a net C accumulation would be observable around those pores. We found that the MAOM mineralization/desorption is the dominating process simply because so little pores in this size range are actually filled with plant residues.

I.311. While I agree that there is no need to repeat in detail the method descriptions that were reported elsewhere, still I believe it is much needed to at least state which methods were used, and possibly provide key method details, for the soil variables that are part of the Results and Discussion in this specific manuscript. Different fractions of POM? WEOC? How all these were measured?

Agreed. We now provide the information in line 473-490.

In addition, here are some other things that I believe would be very beneficial to add to the Methods, and to potentially consider when discussing the results: 1) Depth from which the soil aggregates for the study were collected. 2) The management details, such as tillage and recent crops. 3) Length of time under current land use.

The information is now given in Supplementary Table 1.

Since the study is focusing on soil moisture regimes as the potential driving force for the observed differences, it would be good to provide more information and show some data describing the differences among the regimes.

We do not have additional information other than mean annual precipitation. However, soil types based on diagnostic features can be considered as very strong indication of the soil moisture regime (haplic, stagnic, gleyic).

I.687-688. Maybe separate the portion with POM examples as, say, part d of the figure?

Agreed. POM examples are now moved to the bottom as part d of Fig. 1.

Fig. 2. Methods used for TOC, MAOM, POM, and WEOC measurements need to be listed in the Methods. Also please describe how the share of C in fresh POM was determined.

Done as suggested. (lines 473-490, lines 604-607)

L.702 Is this what pink, blue, green colors represent? Please add a description.

Meaning of colors is now described in the figure caption Also, the color scheme is slightly changed to set it apart from that used for soil types.

I.706. Fig. 4. Can the matrix/POM/dense portions on part b be directly compared to each other? It is very surprising that in Gleysol the POM has lower Os than the matrix.

All materials are subject to the same grayscale normalization. They differ in sub-resolution porosity, which affects the Os intensity, since there can be no Os in voids. Therefore, we do not advocate direct comparison of Os intensities between materials classes. However, in comparison to the other two soils, the material average for soil matrix and POM indicates the high amount of (potentially labile) MAOM and strong C leaching from POM in the Gleysol.

I.711-712 and the Statistical analyses in general (I.453-459). I find it very surprising that such seemingly small differences among the means with such high variability turn out to be all statistically significant. Are you sure that the correct error terms were used in the stats and that the individual POM pieces were not treated as experimental units in this analysis? Please add a description of how the hierarchical nature of the study's experimental design (individual POM pieces, aggregates, soil types) was accommodated in the statistical models used for the data analyses.

In the previous version individual POM pieces were indeed treated as experimental units, which we shouldn't have done and have now corrected. In the revised version, we compute average properties

for individual aggregates (n=8 per soil type) and carry out two-way ANOVA on these aggregate averages (1 factor: soil type, 2. factor POM type). This is now explained in the method section (lines 640-643).

Fig. 6a - I recommend adding more separating lines between different WFPS and anoxic and oxic conditions - to make the figure easier to read. Adding an x-axis line also will be helpful.

Done

Fig. 6b. Not sure I fully understand the measure shown on the y-axis. Is this a ratio of how much C was emitted during anoxic incubation in 1 day divided by the total soil C? Does this mean that in some cases 60% of TOC respired in 1 day?

Please note the unit (mg/g). So less than 1 per mill of TOC was respired per day.

Reviewer #2 (Remarks to the Author):

This study examines the microscale distribution of particulate and mineral associated organic matter (POM and MAOM) in soil types with different drainage regimes. An Osmium (Os) staining approach is used for the first time in combination with a conventional (non-synchrotron) X-ray CT scanner to map the distribution of organic matter within aggregates from each soil. Additional NanoSIMS analyses as well as lab incubations under oxic and anoxic conditions and at different moisture contents are performed to corroborate some of the findings. The authors nicely demonstrate the ability of Os staining and CT imaging to discern fresh and aged POM, with contents varying across the soil types. Os staining within the remaining soil matrix (= everything not considered POM) was defined as MAOM. Os staining also revealed gradients of organic matter around POM fragments as well as pores. Generally, organic matter contents within the soil matrix (as indicated by Os concentration) increased towards POM and decreased around empty pores. The NanoSIMS results seem to indicate that Os stains not just organic matter, but also Fe oxides and clay minerals, which complicates the interpretation of Os as a proxy for organic matter. The incubation results showed a relationship between degree of POM and POM-derived leachate and microbial respiration under anoxic conditions.

This work is an extremely thorough and multifaceted investigation of the 3D distribution of organic matter in soils. How the physical architecture of soils influences soil carbon storage is one of the key research questions in soil science. By combining a unique suite of techniques, this study offers some interesting and novel insights. For example, the fact that visualization and quantification of organic matter distribution around POM and empty pores. The connection between POM leaching and anoxic CO₂ production is also noteworthy.

We would like to extend our gratitude for this positive appraisal of our work.

While I believe this manuscript would be of interest to the broader soil science community, the following points could be clarified/improved upon:

1) The classification of everything that is not POM as MAOM. Given the recent attention the simple, functional classification of organic matter in POM and MAOM pools has received, I understand the desire to partition POM and MAOM in this study. I applaud the authors on this attempt and think it is a useful first step. But I have a hard time with the idea that everything that is not POM in the CT scans can

be classified as MAOM. First of all, the 5 μm voxel size is just too large to truly resolve MAOM, which we would expect to be associated with minerals such as Fe oxides and clays, which are smaller than 2 μm . Second, if Os binds to exactly those minerals, how do we know Os staining signifies MAOM, and not just the presence of these minerals in a certain soil locale? Perhaps it would be more appropriate to present this type of OM as “matrix-bound OM”, which could be truly mineral-associated, but also free amorphous OM, POM smaller than the spatial resolution, or microbial biomass.

We fully agree with the mentioned shortcoming and use “matrix-bound organic matter” or “C in the soil matrix” as alternative terms in the revised manuscript.

2) The assumption that soil moisture is the main driver of the differences in POM/MAOM patterns across the three soils. The three soils don't just differ in the moisture regime, they will also have vastly different C contents, MAOM/POM fraction, C inputs, degrees of bioturbation, mineral composition, etc - all of which will alter the MAOM/POM fraction. I think it is misleading to assume that the observed differentiation in MAOM/POM distribution are solely driven by differences in moisture dynamics. It is not like soil moisture is manipulated, and all other factors remain constant. I don't argue it doesn't play a role, merely that soil structure and the emerging distribution of MAOM/POM within these soils will be a function of the interaction between soil moisture AND these other variables (C inputs, bioturbation, mineralogy, etc.). I think it would help to state more clearly from the beginning that this study compares soils with significantly different characteristics that should result in differences in the spatial arrangement of the organic matter within the pore system.

This comment is in line with concerns of reviewer #1 about the unaccounted role of land use in shaping the microscale C patterns (see comment above). We have now stressed in the Discussion section that MAOM contents, POM contents and various properties tested with the soil incubation experiments are mainly governed by parent material, land use, productivity and so on. We argue that C losses from POM and the microscale C distribution in the soil matrix are the properties that are mainly (but not exclusively) governed by soil moisture regime (see new Fig 8). We also discuss how the C contents around POM could also be explained by different land use instead (line 391-396). As a concluding remark we call for more controlled field studies with the introduced protocol to scrutinize the impact of individual drivers (lines 396-400).

3) Make it clear from the onset why NanoSIMS and incubations were used. Reading through the manuscript, it came as a surprise to see that NanoSIMS, XRF, and additional soil incubations were used, and I wasn't always sure how they were related to the original CT measurements. Perhaps it could be useful to justify in the introduction why this multifaceted approach was used.

We agree and have now added the rationale of additional analyses already in the introduction (lines 72-79).

4) Incubations not conducted under comparable conditions. I'm a bit concerned about the ability to compare the oxic and anoxic incubations if they weren't conducted under the same conditions. Some soil aggregates were embedded, others weren't, but it's unclear why, and it doesn't give me confidence that the experiment really succeeds in isolating O₂ availability as a factor.

We understand your concerns about limited comparability, but can assure that the findings are not compromised by the slightly different setup. For anoxic incubations, soil aggregates were embedded in a silty quartz matrix to protect their structural integrity in line with previous incubation studies

(Wang et al., 2015;Keiluweit et al., 2017). The water saturations were adjusted to 60%WFPS to allow for both optimal gas exchange with the headspace and sufficient water retention within soil pores. The greater oxygen diffusion distances through the silt matrix are irrelevant, since only the anoxic incubation was carried out this way. For the oxic incubations at different water saturations it was important to accurately adjust the WFPS levels in each of the >15 aggregates per incubation vessel, which is why we wetted them individually and placed them loosely into the incubation vessels thereafter. If the presence of the silt matrix would have had an effect on anaerobic respiration, it had been same for all three soils and therefore without an effect on the relative differences in respiration ratios between soils. This statement is now added to the method section (lines 634-637). Finally, the comparison between loosely placed aggregates at 50%WFPS and 100%WFPS is straightforward, as it was carried out under otherwise identical conditions.

5) I had a hard time following the manuscript. There are a lot of different data types and the presentation often wasn't very clear. What I found particularly confusing is the fact that results and discussion aren't fully separated. The many different results are presented bit by bit, but then discussed individually. In the actual discussion, these many different pieces aren't really brought together and synthesized. So, when the conceptual model is presented, it's not very clear where that is coming from. I think it would help if the results were more compact and to the point, and were then followed by a discussion that (i) synthesizes the results and (ii) articulates how these results informed the conceptual model that was developed.

We have now attempted to completely disentangle Results and Discussion. The Discussion starts now with an appraisal of the novel protocol, then addresses the microscale observations, and finally the ramifications for C turnover. The new structure of the Discussion is underpinned by a completely revised conceptual model that integrates all findings, including the incubation results (see new Fig. 8).

Given those limitations, I wonder if there is a story the authors could develop around the visualization of POM, its decomposition and penetration of the surrounding soil matrix, and its role in mediating C mineralization under varying O₂/moisture contents. That would be a very interesting story (to me) that could be told with without explicitly mentioning MAOM.

We hope that the suggested story line is now carved out better. More emphasis is now put on the C losses from POM and incorporation of POM into the soil in response to structure turnover.

Abstract

19: What is the "thickness of MAOM depletion"? Can depletion have a thickness? Also, depletion relative to what? Are you saying the layers become less thick? Finally, what do you mean by "around pores"? Any pores as measured by X-ray CT, micropores, macropores?

We have now completely rewritten the second part of the abstract according to the suggested story line. More specifically, around pores" is replaced by "around pores (aperture >10 μm)".

Intro

29-34: If that's already know, what is really the knowledge gap explored here?

The knowledge gap is that the resulting microscale organic carbon patterns are unknown, which are formed by the interplay of the listed processes under natural conditions. This is now clearly expressed at the end of the paragraph. (line 50-52)

Hypothesis seems a bit like a strawman – is it truly falsifiable?

Agreed. We have rephrased the hypothesis in a way that it can be falsified: “Based on these macroscopic observations we hypothesize that soil moisture regimes shape characteristic C distribution patterns at the pore scale.” (line 53-56)

56-59: Seems like the actual insight gained from this study. Maybe emphasize more?

Agreed. We now emphasize it more by stressing that such a comparison has been carried out for the first time. (Lines 66-67)

66-72: These soils will differ in mineralogy, structure, C inputs, etc – IE more than one factor (soil moisture regime) varies, which makes it difficult to attribute the observations solely to variations in soil moisture. It would be more appropriate to state clearly (e.g. in the abstract) that soils across a drainage gradient were examined and showed the pattern reported here.

Please consider that the word count for the abstract is very restrictive. Instead of drainage gradient we refer to three different soil types with fundamentally different soil moisture regimes (well aerated, stagnic, and gleyic). “With three different soil types we show that the soil moisture regime governs C leaching from POM and the microscale C redistribution and stabilization patterns in the soil matrix” (line 23-25). A full discussion of other factors that may drive C turnover and C pattern formation follows in the discussion section.

Fig. 1: The text talks a fair amount of MAOM identified in Fig. 1, but I can't see anything labeled as MAOM here. Please explain. Also, how do you know the nodules are Fe-rich?

The green material class is labeled as matrix. Therefore all OS_{CT} intensities within this material class represent matrix-bound organic matter. A material legend is now added to Fig. 1, so that this information does not have to be retrieved from the caption. The confirmation of high Fe contents in these nodules using μ XRF and NanoSIMS follows later.

100: abbreviate as POM?

Done.

105-107: X-ray CT quantification of POM seems good (significant), but that of MAOM is not great (not significant) (Fig. 2). Makes sense because it's just quantified by subtraction. Maybe only highlight POM, which can be quantified with higher confidence?

No, we want to show this poorer match between image-derived, matrix-associated OM and MAOM on purpose to 1) highlight the shortcomings of limited voxel resolution and 2) to show that there is substantial Os sorption to the matrix even at vanishing MAOM content.

119: “perfectly”? Maybe a bit much with three data points.

Replaced by “adequately” (line 138).

127-129: Isn't the fact that Os binds to exactly those minerals that are principally responsible for MAOM formation (Fe oxides and clay minerals) a problem? How do you know if it binds because of the affinity for the mineral as opposed to the presence of MAOM?

Yes, this is a problem. Using Os staining and X-ray CT alone is not sufficient to disentangle both binding mechanisms. It is now clearly identified as such (Lines 172-176)

132-138: Ok, so I understand you're trying to use NanoSIMS to resolve that. But NanoSIMS just seems to reinforce that point, i.e., Os binding to clay minerals and Fe oxides? How do you know it's MAOM?

We cannot know for sure. This is what clearly state below

143: Point well taken

147: Why can relative differences among soils still be interpreted? What if one soil has more Fe oxides/clays than the other? That would lead to larger “MAOM” estimates even if there is not OM bound to the minerals due to the affinity of Os to the minerals. I'm sure you've thought this through, but I'm not following ...

We admit this rationale should have been explained better. In fact, we advise against comparisons between soils of different mineralogy. “Among soils” should have been replaced by “within soils”. We argue that “average matrix-bound organic matter contents can still be investigated with respect to pore or POM distances assuming that mineralogy is not changed systematically at the studied microscale distances” (lines 290-294). This assumption is then put to test with μ XRF data (line 295-305).

176: it would be useful to know on what grounds that possibility was ruled out

It was ruled out based on X-ray CT scans at different spatial resolutions (13 vs. 2 μ m). Information now added to the text (line 306-307).

Fig. S5: It would help to see how well the signal intensity is correlated. The maps don't seem that similar and there are hotspots in both maps that only show in one or the other.

We would not know how to interpret an individual correlation coefficient since it cannot be compared to anything else. In addition, we know fairly well the reasons for the poor correlation (different lateral and depth resolution).

177ff: isn't it possible that this is an artifact of the X-ray CT technique and where the threshold is drawn for the segmentation? The boundary between “pore” and “matrix” might not always be super clean, right? And, consequently, the density of the material surrounding the pores might be less dense and thus bind less Os? I don't know much about X-ray CT, but the fact that this “depletion” occurs right at the segmentation threshold that is always tricky to get right raises some questions for me.

We appreciate that the reviewer has put quite some thought into the admittedly delicate task of image segmentation. Yes, there are always partial volume voxels at material interfaces. In contrast to simple thresholding, the machine learning classifier is also trained on gradient information and texture information to identify material boundaries particularly well. Most of the training is, in fact, carried out by drawing thin lines for both materials at both sides of a few material interfaces. Partial volume voxels at the material interface between macropores and soil matrix would indeed have an extent of 1-2 voxels into the soil matrix. This can be shown by depicting the Os gradient from the material boundary into macropores. This extent is much shorter than the 6-16 voxels (30-80 μm) and beyond before a plateau in matrix-bound Os signal is reached.

189 and 192: these statements are examples where I think focusing on the results without further discussion would be helpful. These statements clearly belong in the discussion and, in the present form, don't feel very well supported and appear speculative.

Agreed. All speculative statements are moved to the Discussion section.

Fig. 5a. I suggest labeling the box plots (b or c) as well so you can refer to them in the text more easily. Otherwise, it remains difficult to assess which aspect of 5a is discussed in the text.

Done.

205: Is it possible that Fe and Al are less abundant in the vicinity of POM in the gleysol and that's why you are observing less Os binding? It would seem that POM would be a hotspots for leaching of both Al and Fe due to pH or redox driven leaching of both.

This is an interesting thought, but unfortunately difficult to implement technically. POM is always surrounded by pores. We know that they exist from X-ray CT images (voxel size: 5 μm), but they are not resolved by the coarser μXRF solution (spot size: 20 μm , depth resolution $>100 \mu\text{m}$). So if there was Al and Fe leaching around POM, we could not separate it from sub-resolution porosity gradients around POM.

Section 2.3 header: I would suggest making it clear this is a results section. "implications..." made me think this is part of the discussion.

Agreed. Implications has been removed from the header.

221: p value is different from what is shown in Fig. 6b. Also 6b only shows fresh POM, so this sentence is a bit confusing.

We are sorry for the inconsistent p-value and axis label, which have now been corrected in the main text (Line 253-255).

Discussion

I believe that this paper would really benefit from a concise summary of the results before the presentation of the emerging conceptual model. There are a lot of results to digest, and it would be

helpful to the reader to know how these data come together to paint the picture you are presenting in the following.

Agreed. We have revised the Discussion section and the conceptual figure accordingly (see response above).

240-252: This conceptualization disregards the role that differences in the mineralogical composition of the soils might play in the formation of MAOM in these soils. This overview also doesn't take into account the incubation results, which reinforces the disconnect between the imaging results and the incubation.

If mineralogical composition mattered, it would most likely do so everywhere in the soil matrix and therefore not evoke in spatial Os gradients after averaging Os intensities with respect to POM and pores. The link between incubation and imaging results is the different shares of C stored in the two POM types and not the spatial gradients. Therefore the conceptual model has been extended in this regard (Fig 8). The synoptic overview clearly separates now between properties that are mainly governed by soil moisture regime (C losses from POM and microscale gradients in matrix-bound OM) and other properties (POM and MAOM content, bioavailability, biodegradability) that are influenced by other factors, such as mineralogy, land use, etc.

284: This is the conclusion/summary paragraph. Yet, I believe this is the first time biopores are mentioned. Another complicated concept that needs more introduction or should be omitted to avoid further complicating the story.

Agreed. The term biopores is replaced by POM-bearing pores.

437: WHC reported here, but WFPS reported in Fig. 6. Which one is it?

445: How does 40 or 60% WFPS compare to the WHC numbers reported above?

This was just a typing mistake. WHC has now been corrected to WFPS.

446: On what basis? The quartz silt mixture is going to hold some water and influence gas transport, so alter the dynamics in the soil aggregates. I don't understand why the oxic and anoxic incubations weren't performed under the same conditions if they are to be compared. Some justification or validation of the assumption that these treatments can be compared is necessary.

Agreed (See comments above). We have now explained in more detail why the presence of a silt matrix is considered to be irrelevant under anoxic conditions (L. 634-637).

References

- Elyeznasni, N., Sellami, F., Pot, V., Benoit, P., Vieubl e-Gonod, L., Young, I., and Peth, S.: Exploration of soil micromorphology to identify coarse-sized OM assemblages in X-ray CT images of undisturbed cultivated soil cores, *Geoderma*, 179-180, 38-45, <https://doi.org/10.1016/j.geoderma.2012.02.023>, 2012.
- Fimmen, R. L., Richter, D. d., Vasudevan, D., Williams, M. A., and West, L. T.: Rhizogenic Fe-C redox cycling: a hypothetical biogeochemical mechanism that drives crustal weathering in upland soils, *Biogeochemistry*, 87, 127-141, 10.1007/s10533-007-9172-5, 2008.

Grybos, M., Davranche, M., Gruau, G., Petitjean, P., and Pédrot, M.: Increasing pH drives organic matter solubilization from wetland soils under reducing conditions, *Geoderma*, 154, 13-19, <https://doi.org/10.1016/j.geoderma.2009.09.001>, 2009.

Hagedorn, F., Kaiser, K., Feyen, H., and Schlegli, P.: Effects of Redox Conditions and Flow Processes on the Mobility of Dissolved Organic Carbon and Nitrogen in a Forest Soil, *Journal of Environmental Quality*, 29, 288-297, <https://doi.org/10.2134/jeq2000.00472425002900010036x>, 2000.

Keiluweit, M., Wanzek, T., Kleber, M., Nico, P., and Fendorf, S.: Anaerobic microsites have an unaccounted role in soil carbon stabilization, *Nature Communications*, 8, 1771, 10.1038/s41467-017-01406-6, 2017.

Kravchenko, A. N., Guber, A. K., Razavi, B. S., Koestel, J., Quigley, M. Y., Robertson, G. P., and Kuzyakov, Y.: Microbial spatial footprint as a driver of soil carbon stabilization, *Nature Communications*, 10, 3121, 10.1038/s41467-019-11057-4, 2019.

Schulz, M., Stonestrom, D., Lawrence, C., Bullen, T., Fitzpatrick, J., Kyker-Snowman, E., Manning, J., and Mních, M.: Structured Heterogeneity in a Marine Terrace Chronosequence: Upland Mottling, *Vadose Zone Journal*, 15, 10.2136/vzj2015.07.0102, 2016.

Wang, Q., Wang, D., Wen, X., Yu, G., He, N., and Wang, R.: Differences in SOM Decomposition and Temperature Sensitivity among Soil Aggregate Size Classes in a Temperate Grasslands, *PLOS ONE*, 10, e0117033, 10.1371/journal.pone.0117033, 2015.

REVIEWERS' COMMENTS

Reviewer #2 (Remarks to the Author):

The authors addressed my comments comprehensively. I read through it and have only the following comments in regards to general readability:

22-24: I have a hard time understanding what this means. How can C depletion “account” for a larger soil volume? Do you mean a larger soil volume is characterized by C depletion?

53-60: I really like this section

70-72: I’m not sure I understand. Why can they be investigated if they don’t change systematically and are highly variable? Shouldn’t it be the other way round?

175-180: There is a funny text segment where the punctuation doesn’t seem quite right and the font is off.

271: I suggest talking about organic matter “pools” instead of “fractions” unless you are referring to operational procedures.

273: Abbrev. as POM

Dear Editor and Reviewers,

thank you once again for a thorough review and good suggestions. Please find our responses to each comment below. Your comments are reproduced in normal font, our response is in bold font.

We therefore invite you to revise your paper one last time to address the remaining concerns of our reviewers and our editorial requests in the attached document(s). As you will see from the reports, Reviewer #1 was not able to supply a review this time. Reviewer #2 assessed the extent to which Reviewer #1's concerns were addressed, and while they were mostly satisfied, they did agree with Reviewer #1's previous comment about the "fresh" versus "aged" distinction and did not feel that such a distinction was well supported. Reviewer #2 suggested using terms that describe the morphological characteristics of the POM type, rather than the suggestive terms that imply a certain decomposition stage.

We think that the interpretation of morphological properties towards decomposition stage is justified, the more so as it is based on previous, independent work. In order to comply with the suggestion of reviewer #2 we will use the morphological terms “fibrous” and “compact” POM in the method and result section, but pick the topic up again in the discussion section as to how far it can be interpreted as decomposition stage.

Additionally, we ask that you edit your manuscript to comply with our policies and formatting requirements and to maximise the accessibility and therefore the impact of your work. Please see the attached document(s), listing a number of points that must be addressed. Failure to comply with our editorial requests will cause delays in accepting your manuscript. Please also see the *Nature Communications* formatting instructions for further information.

We addressed all points in the checklist.

Reviewer #2 (Remarks to the Author):

The authors addressed my comments comprehensively. I read through it and have only the following comments in regards to general readability:

22-24: I have a hard time understanding what this means. How can C depletion “account” for a larger soil volume? Do you mean a larger soil volume is characterized by C depletion?

Agreed. We rephrased the respective sentence: “Carbon depletion around pores (aperture >10 µm) occurs in a much larger soil volume (19-74%) than C enrichment around POM (1%).”

53-60: I really like this section

Thank you.

70-72: I'm not sure I understand. Why can they be investigated if they don't change systematically and are highly variable? Shouldn't it be the other way round?

This is a misunderstanding. Hopefully it is clearer after re-phrasing the statement: "Relative changes in average Os sorption as a function of pore or POM distances can still be interpreted as spatial gradients in sorption to organic matter assuming that Os sorption to mineral surfaces is random with respect to these distances."

175-180: There is a funny text segment where the punctuation doesn't seem quite right and the font is off.

The sentences have been simplified. Information in parenthesis has been removed or is now described in sentences. The strange font appeared only after conversion from word to PDF. This problem has been solved now.

271: I suggest talking about organic matter "pools" instead of "fractions" unless you are referring to operational procedures.

Done.

273: Abbrev. as POM

Done.